# FIRE: Frobenius-Isometry Reinitialization for Balancing the Stability–Plasticity Tradeoff

**Isaac Han**[1]      **Sangyeon Park**[1]      **Seungwon Oh**[1]      **Donghu Kim**[2,3]

**Hojoon Lee**[2,3]*      **Kyung-Joong Kim**[1]*

[1]Gwangju Institute of Science and Technology (GIST)
[2]Korea Advanced Institute of Science and Technology (KAIST)
[3]Holiday Robotics
* Equal advising
{lssac7778, tkddus0421, sw980907}@gm.gist.ac.kr

## Abstract

Deep neural networks trained on nonstationary data must balance stability (i.e., retaining prior knowledge) and plasticity (i.e., adapting to new tasks). Standard reinitialization methods, which reinitialize weights toward their original values, are widely used but difficult to tune: conservative reinitializations fail to restore plasticity, while aggressive ones erase useful knowledge. We propose FIRE, a principled reinitialization method that explicitly balances the stability–plasticity tradeoff. FIRE quantifies stability through Squared Frobenius Error (SFE), measuring proximity to past weights, and plasticity through Deviation from Isometry (DfI), reflecting weight isotropy. The reinitialization point is obtained by solving a constrained optimization problem, minimizing SFE subject to DfI being zero, which is efficiently approximated by Newton–Schulz iteration. FIRE is evaluated on continual visual learning (CIFAR-10 with ResNet-18), language modeling (OpenWebText with GPT-0.1B), and reinforcement learning (HumanoidBench with SAC and Atari games with DQN). Across all domains, FIRE consistently outperforms both naive training without intervention and standard reinitialization methods, demonstrating effective balancing of the stability–plasticity tradeoff. Explore codes and videos at project page: https://isaac7778.github.io/fire/.

## 1 Introduction

Deep neural networks are typically trained under a fixed, stationary data distribution (Brown et al., 2020; Podell et al., 2024). However, many real-world applications require models to adapt continually as new data and shifting distributions emerge. In computer vision, autonomous driving systems must recognize unseen traffic signs, road layouts, or weather conditions that were absent during training (Verwimp et al., 2023). Large language models, trained once and deployed with a fixed knowledge cutoff date, quickly become outdated unless continually updated (Ke et al., 2023). Likewise, robots deployed in dynamic physical environments must adjust their perception and control policies as the environment changes (Wołczyk et al., 2021). In all of these domains, a central challenge is reliable adaptation to nonstationary data while preserving prior knowledge.

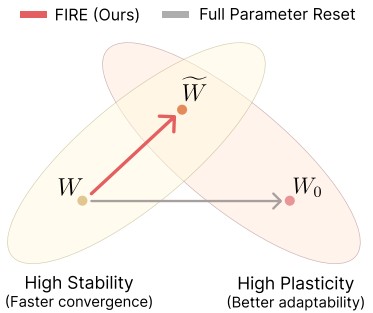

Figure 1: **Illustration of FIRE.** Solving a constrained optimization problem, FIRE places weights at the intersection of high-stability and high-plasticity manifolds.

This challenge is often framed as a balance between two competing properties: *stability*, the retention of learned knowledge, and *plasticity*, the ability to incorporate new information (Mermillod et al., 2013). Different research communities emphasize these properties to varying degrees. Conventional

continual learning assumes limited access to past data, prioritizing stability to mitigate catastrophic forgetting (Kirkpatrick et al., 2017; Rebuffi et al., 2017; Rusu et al., 2016). In contrast, most foundation models and robotic agents are trained on expanding datasets where past data remain accessible (Achiam et al., 2023; Team et al., 2025), making plasticity loss a central challenge (Lyle et al., 2023; Berariu et al., 2021). In this regime, stability is less about preserving past knowledge and more about accelerating adaptation to new tasks by leveraging prior representations. Motivated by these real world scenarios, we study the stability-plasticity tradeoff under the assumption of access to past data during continual learning.

Existing approaches to mitigating plasticity loss fall broadly into two categories: regularization-based and reinitialization-based. Regularization-based methods constrain parameters or features near their initialization (Kumar et al., 2025b; Lyle et al., 2022), or enforce weight orthogonality (Chung et al., 2024). While these methods can preserve a favorable geometry for future learning, overly strong constraints slow convergence, and overly weak ones fail to prevent plasticity degradation. Reinitialization-based methods instead reset weights to earlier checkpoints when new data arrive (Ash & Adams, 2020; Nikishin et al., 2022; Lee et al., 2024a; Shin et al., 2024). Their advantage lies in avoiding interference with current optimization, often yielding faster adaptation with lower overhead. However, they also suffer from a tuning dilemma: aggressive resets erase useful knowledge, while conservative ones provide little plasticity gain.

We aim to resolve this dilemma by treating reinitialization as a principled constrained optimization problem. Our approach relies on two complementary measures that capture the core dimensions of the stability–plasticity tradeoff. First, we define stability using the Squared Frobenius Error (SFE) between current and past weights, which measures the sum of squared differences across all weight entries. A smaller SFE indicates greater similarity, meaning the model remains closer to its previous representations. For plasticity, prior work has linked plasticity loss to sharp loss curvature (Lyle et al., 2023), dormant neurons (Sokar et al., 2023), and low rank features (Kumar et al., 2021a), but these metrics depend on incoming data and are non differentiable, limiting their use for optimization. We instead propose the Deviation from Isometry (DfI) (Pennington et al., 2017; Xiao et al., 2018), which measures how close weight matrices are to orthonormal. We show that reducing DfI simultaneously decreases curvature, prevents neuron dormancy, increases feature rank, and remains differentiable, making it a practical measure of plasticity to optimize. A formal proof is provided in Section 3.

We propose FIRE (Frobenius–Isometry REinitialization), which minimizes the SFE subject to the DfI being 0. As illustrated in Figure 1, FIRE avoids the pitfalls of either overly conservative or aggressive reinitialization by projecting weights onto the isotropy manifold while remaining close to their previous subspace. While directly solving this constrained optimization is costly, it can be implemented efficiently with the Newton–Schulz iteration, adding less than 1% to training time.

We evaluate FIRE on continual learning benchmarks in vision, language, and reinforcement learning, assuming access to past data. For vision, we split CIFAR-10, CIFAR-100, and Tiny-ImageNet into chunks under random or class-incremental protocols using ResNet (He et al., 2016) and Vision Transformer (Dosovitskiy et al., 2020). For language, we use a warm-start setup where GPT-0.1B (Karpathy, 2023) is pretrained on WikiText-103 and then continually trained on a mixture of OpenWebText and WikiText-103. For reinforcement learning, we test continuous control with SAC (Haarnoja et al., 2018) on HumanoidBench (Sferrazza et al., 2024) and discrete control with DQN (Mnih et al., 2015) on Atari (Bellemare et al., 2013). For vision and language tasks, reinitialization is applied whenever new data arrive, while in reinforcement learning it is applied once at the midpoint of training. Across all domains, FIRE consistently outperforms naive training and standard reinitialization, showing its effectiveness as a unified solution to the stability plasticity tradeoff.

## 2 RELATED WORK

### 2.1 STABILITY-PLASTICITY TRADEOFF

The stability–plasticity tradeoff (Mermillod et al., 2013; Kim & Han, 2023) is a fundamental challenge in continual learning. Stability refers to the ability of a model to preserve previously acquired knowledge and avoid catastrophic forgetting when exposed to new data. Plasticity refers to the ability of a model to adapt flexibly and effectively to novel tasks. These two properties often conflict

with each other since strong stability can make the model rigid and resistant to new learning while excessive plasticity can lead to the loss of past knowledge.

Research in continual learning has therefore focused on methods that balance these competing requirements such as constraining parameter updates through regularization (Kirkpatrick et al., 2017), revisiting earlier data through replay (Rebuffi et al., 2017; Rolnick et al., 2019; Lopez-Paz & Ranzato, 2017; Chaudhry et al., 2018; Aljundi et al., 2019), or designing architectures that separate parameters across tasks (Rusu et al., 2016; Mallya & Lazebnik, 2018; Mallya et al., 2018; Yoon et al., 2017; Wortsman et al., 2020). Their aim is to develop models that can maintain previously learned skills while remaining adaptive to new experiences.

## 2.2 Loss of plasticity

Deep learning has traditionally been studied under stationary datasets (Glorot & Bengio, 2010; He et al., 2015), yet real-world applications often involve non-stationary streams (Shen et al., 2024; Kumar et al., 2025a). Training in such environments leads to a loss of plasticity (Lyle et al., 2023; Dohare et al., 2024; Kumar et al., 2025b), where models fail to adapt to new distributions. Prior work has identified potential indicators of this phenomenon, including dormant neurons (Sokar et al., 2023; Xu et al., 2023), shifts in pre-activations (Lyle et al., 2024), feature rank collapse (Kumar et al., 2021a), and diverging weight magnitudes (Lyle et al., 2024).

Loss of plasticity hinders not only the ability to fit the training data, but also the ability to generalize to unseen data. Models trained incrementally often generalize worse than those trained from scratch (Ash & Adams, 2020; Berariu et al., 2021; Lyle et al., 2025), due to factors such as diminished gradient norms (Ash & Adams, 2020), weak feature changes (Lyle et al., 2025), and the compounding effects of small pretraining datasets or noisy labels (Lee et al., 2024a).

To counteract plasticity loss, reinitialization-based strategies such as S&P (Ash & Adams, 2020), DASH (Shin et al., 2024) reinitialize weights into an intermediate checkpoint, weight regularizers constrain parameters to initialization or specific subspaces (Kumar et al., 2025b; Elsayed et al., 2024; Lewandowski et al., 2024a), and spectral or rank-based approaches explicitly maintain representation quality (Kumar et al., 2021a;b; He et al., 2024). Another approach proposed reinitializing at the neuronal level, based on the utility of each neuron (Sokar et al., 2023; Dohare et al., 2024; Elsayed & Mahmood, 2024). Recent work further leverages the fact that linear networks do not suffer from plasticity loss (Dohare et al., 2024; Lewandowski et al., 2024b; Park et al., 2025).

## 3 Method

In this section we explain how we frame the stability-plasticity tradeoff as a constrained optimization problem. To do this we first need two metrics: one for stability and one for plasticity loss. With these two pieces in place the optimization problem naturally emerges, and from this formulation we introduce our method FIRE, which provides an efficient approximation to the solution.

## 3.1 Measure for stability

To measure stability, we propose a simple yet effective metric, the *Squared Frobenius Error (SFE)*. SFE provides a natural way to quantify the preservation of learned information by comparing an original weight matrix $W$ with its modified counterpart $\widetilde{W}$. Formally,

$$\text{SFE}(W, \widetilde{W}) = \|W - \widetilde{W}\|_F^2. \tag{1}$$

which captures the element-wise squared deviation between the two weight configurations. However, it remains unclear whether SFE can be used as a metric that can meaningfully capture similarity of feature representations. To clarify this point, we establish a theoretical link between SFE and the normalized feature covariance, a metric widely used in prior work to measure representation similarity (Lyle et al., 2025; Yang et al., 2022). In particular, we show that SFE provides an upper bound on the discrepancy between the normalized feature covariances of two distinct neural networks' output features (Theorem 1).

**Theorem 1** (SFE bounds output feature covariance between two deep neural networks). *Let $\Theta = \{W^1, \ldots, W^L\}$ and $\widetilde{\Theta} = \{\widetilde{W}^1, \ldots, \widetilde{W}^L\}$ be the parameters of two depth-L feedforward networks with elementwise activations $\sigma_\ell$ (Lipschitz constants $L_{\sigma_\ell}$).*

*For an input batch $Z \in \mathbb{R}^{n \times d_0}$, we denote the layer outputs recursively by $H^0_\Theta(Z) = Z$ and $H^\ell_\Theta(Z) = \sigma_\ell(H^{\ell-1}_\Theta(Z)W^\ell)$. Let $B_\ell = \max\{\|W^\ell\|_2, \|\widetilde{W}^\ell\|_2\}$ be the maximum spectral norm of the weights in layer $\ell$ and $B^\ell_\Pi = \prod_{k=1}^\ell B_k$ the product across all layers. We further define $m_\ell = \min\{\|H^\ell_\Theta(Z)\|_F, \|H^\ell_{\widetilde{\Theta}}(Z)\|_F\} > 0$. The normalized feature covariances of the two networks are given by $C^\ell_\Theta(Z) = H^\ell_\Theta(Z)H^\ell_\Theta(Z)^\top / \|H^\ell_\Theta(Z)\|^2_F$, $\quad C^\ell_{\widetilde{\Theta}}(Z) = H^\ell_{\widetilde{\Theta}}(Z)H^\ell_{\widetilde{\Theta}}(Z)^\top / \|H^\ell_{\widetilde{\Theta}}(Z)\|^2_F$.*

*Then the difference between the output feature covariances of the two networks is bounded as follows:*

$$\|C^\ell_\Theta(Z) - C^\ell_{\widetilde{\Theta}}(Z)\|^2_F \leq \frac{16 \, \|Z\|^2_F}{m^2_\ell} \Big( \prod_{k=1}^\ell L_{\sigma_k} \Big)^2 B^2_\Pi \Big( \sum_{j=1}^\ell \tfrac{1}{B^2_j} \Big) \mathrm{SFE}(\Theta, \widetilde{\Theta}).$$

*In particular, if each activation is 1-Lipschitz $(L_{\sigma_k} \leq 1)$ and each spectral norm is bounded $(B_j \leq S)$, then*

$$\|C^\ell_\Theta(Z) - C^\ell_{\widetilde{\Theta}}(Z)\|_F \ \leq \ \frac{4 \, \|Z\|_F}{m_\ell} \, \sqrt{\ell} \, S^{\ell-1} \, \sqrt{\mathrm{SFE}(\Theta, \widetilde{\Theta})}.$$

Theorem 1 shows that the discrepancy between normalized feature covariances is bounded by four factors: the input norm ($\|Z\|^2_F$), the Lipschitz constants of activation functions ($L_{\sigma_\ell}$), the spectral norms of the weight matrices ($B_\Pi$ and $B_j$), and SFE. In practice, inputs are typically normalized and have a fixed upper bound, and commonly used activation functions have small Lipschitz constants (e.g., 1 for ReLU and Tanh). Thus, the contributions from $\|Z\|^2_F$ and $L_{\sigma_\ell}$ can be ignored. The remaining dominant factors are the spectral norms of the weight matrices ($B_\Pi$ and $B_j$) and SFE. This has two implications. First, for any fixed architecture and weight scale, minimizing SFE monotonically tightens the upper bound, and thus is an effective way to preserve feature similarity between two networks. Second, the slope of the bound with respect to SFE is proportional to the spectral norms: larger spectral norms make the worst-case discrepancy more sensitive to changes in SFE. In such regimes, reducing SFE becomes even more important for stability, since even a small increase in SFE can, in principle, lead to a substantial change in the resulting feature representations.

## 3.2 Measure for plasticity loss

In this work, we view plasticity loss as the situation where weights learned from previous data fail to serve as a favorable initialization point for new data. This perspective is particularly useful when designing reinitialization strategies that partially reinitialize weights before the arrival of new data. Prior analyses of plasticity loss suggest that one of its main causes is the increasing sharpness of the loss landscape curvature with respect to new data during training on the current task, which destabilizes optimization (Lyle et al., 2023). In addition, an increase in dormant neurons (Sokar et al., 2023) and a collapse in effective rank (Kumar et al., 2021a) have also been known to indicate plasticity loss.

However, as these measures are data-dependent and non-differentiable, they are inappropriate for the optimization objective. Instead, we propose Deviation from Isometry (DfI) (Pennington et al., 2017; Xiao et al., 2018), an optimizable metric that closely connected to previous plasticity measure.

$$\mathrm{DfI}(W) = \|W^\top W - I\|^2_F. \tag{2}$$

Our theoretical analysis reveals that minimizing DfI results in a smoother loss landscape curvature (Theorem 2), a smaller number of dormant neurons (Theorem 4), and a higher effective rank (Theorem 3). This supports the use of DfI as a suitable measure for plasticity measure. In addition, minimizing DfI also makes the weights close to isotropy, which is known as a key property of favorable neural network initializations, termed as dynamical isometry (Xiao et al., 2018).

**Theorem 2** (Hessian spectral norm bounded by layerwise DfIs). *We assume that*

- *the inputs are whitened, so that the empirical covariance $\Sigma_Z = \frac{1}{n}Z^\top Z$ is approximately the identity: $\Sigma_Z := \frac{1}{n}Z^\top Z \approx I$*

- *for every sample $i$ and relevant parameter vector $u$, the Hessian norm satisfies $|\nabla_u^2 \ell_i(u)|_2 \le \beta$, while the gradient norm is bounded by $|\nabla_u \ell_i(u)|_2 \le \gamma$.*

- *finally, we focus on a fixed ReLU activation pattern at the point of interest, so that the network is piecewise linear in that region. In particular, each diagonal gating matrix arising from the ReLU has operator norm $\le 1$.*

*Let $\nu_k = 1 + \sqrt{\mathrm{DfI}(W_k)}$, then the Hessian spectral norm can be bounded as follows:*

$$\left\| \nabla_\theta^2 \mathcal{L}(W_{1:L}) \right\|_2 \ \le \ \beta \sum_{k=1}^{L} \prod_{j \ne k} \nu_j \ + \ 2\gamma \sum_{1 \le k < \ell \le L} \prod_{j \notin \{k,\ell\}} \nu_j.$$

**Theorem 3** (DfI controls effective rank). *Let $Z \in \mathbb{R}^{n \times a}$ and $W \in \mathbb{R}^{a \times b}$ be an input and weight matrix, respectively, $\Phi = ZW$ be a feature matrix. Let $\Sigma_Z = \frac{1}{n} Z^\top Z$ be the empirical covariance of the inputs and let $W = QS$ be the right polar decomposition of $W$, where $Q \in \mathbb{R}^{a \times b}$ has orthonormal columns. Then $\eta_1 \ge \cdots \ge \eta_d > 0$ denote the positive eigenvalues of $Q^\top \Sigma_Z Q$, with $d = \mathrm{rank}(Q^\top \Sigma_Z Q)$, and $\mathrm{srank}_\delta(\Phi)$ be defined from the nonzero singular values $\{\sigma_i(\Phi)\}_{i=1}^d$ by $\mathrm{srank}_\delta(\Phi) = \min \left\{ k : \left( \sum_{i=1}^k \sigma_i(\Phi) / \sum_{i=1}^d \sigma_i(\Phi) \right) \ge 1 - \delta \right\}.$*

*Then $\varepsilon = \sqrt{\mathrm{DfI}(W)} < 1$ gives a lower bound on the srank as left inequality below. If additionally $\Sigma_Z = I$, the bound simplifies to the right inequality below.*

$$\mathrm{srank}_\delta(\Phi) \ \ge \ \left\lceil \frac{(1-\delta)\,d}{\delta\,\sqrt{\frac{1+\varepsilon}{1-\varepsilon}}\sqrt{\frac{\eta_1}{\eta_d}} + (1-\delta)} \right\rceil, \qquad \mathrm{srank}_\delta(\Phi) \ \ge \ \left\lceil \frac{(1-\delta)\,d}{\delta\,\sqrt{\frac{1+\varepsilon}{1-\varepsilon}} + (1-\delta)} \right\rceil. \quad (3)$$

**Theorem 4** (Minimizing DfI increases neuron activity score). *Let $W \in \mathbb{R}^{a \times b}$ and let $\sigma$ be positive-homogeneous ($\sigma(\alpha t) = \alpha \sigma(t)$ for $\alpha \ge 0$). Assume the input vector $z \sim \mathcal{N}(0, I_a)$ (isotropic Gaussian). For neuron $j$ with column $w_j$ of $W$, define activity score of neuron $j$ as $s_j = \frac{\mathbb{E}_z[|\sigma(\langle z, w_j \rangle)|]}{\frac{1}{b}\sum_{k=1}^b \mathbb{E}_z[|\sigma(\langle z, w_k \rangle)|]}$. If $\varepsilon := \sqrt{\mathrm{DfI}(W)} < 1$, then for all $j$,*

$$\sqrt{\frac{1-\varepsilon}{1+\varepsilon}} \ \le \ s_j \ \le \ \sqrt{\frac{1+\varepsilon}{1-\varepsilon}}.$$

Sokar et al. (2023) classified neurons with $s_j < \tau$ as dormant neurons. Hence, reducing the number of dormant neurons requires increasing the activity scores $s_j$. Theorem 4 states that minimizing DfI increases the lower bound $\sqrt{\frac{1-\epsilon}{1+\epsilon}} < s_j$. Note that minimizing DfI also decreases the corresponding upper bound. This is particularly meaningful because the neuron activity score $s_j$ is a relative measure, defined as an activation normalized by the mean activation within the same layer. Consequently, it is impossible to increase all $s_j$ simultaneously, since they are normalized by their average. To reduce dormant neurons, it is therefore critical to reduce the discrepancy in activations across neurons rather than uniformly scaling them. Theorem 4 supports this perspective: minimizing DfI tightens both the lower and upper bounds on $s_j$, thereby limiting the score discrepancy between neurons and effectively reduces dormant neurons.

We provided detailed proofs in Appendix A. Theorem 2, 3, 4 demonstrates that reducing DfI is directly associated with maintaining a smoother loss landscape, a high effective rank, and large number of active units.

### 3.3 BALANCING BETWEEN STABILITY AND PLASTICITY

To achieve low DfI while minimizing the loss of information, we formulate the problem as a constrained optimization. Specifically, we minimize the SFE between the original weights $W$ and their orthogonalized counterpart $\widetilde{W}$, subject to the orthogonality constraint $\widetilde{W}^\top \widetilde{W} = I$, which is equivalent to requiring $\mathrm{DfI}(\widetilde{W}) = \| \widetilde{W}^\top \widetilde{W} - I \|_F^2 = 0$.

$$\min_{\widetilde{W}} \ \| W - \widetilde{W} \|_F^2 \quad \text{s.t.} \quad \widetilde{W}^\top \widetilde{W} = I. \qquad (4)$$

This formulation is mathematically equivalent to the well-studied *Orthogonal Procrustes Problem* (Schönemann, 1966), whose solution can be expressed in closed form via the polar decomposition:

$$\widetilde{W}^{\star} = W\left(W^{\top}W\right)^{-\frac{1}{2}}. \tag{5}$$

While the optimization itself is classical, our contribution lies in leveraging this operation as a principled mechanism to balance stability and plasticity in neural networks. In particular, we reinterpret equation 5 as a projection that simultaneously drives the spectrum of $W$ toward isotropy (low DfI) while maintaining stability (low SFE). We provide a derivation of Equation 5 in the Appendix A.

### 3.4 APPROXIMATING THE SOLUTION

Directly computing $\widetilde{W}^{\star}$ exactly can be expensive, we efficiently approximate it using the Newton–Schulz iteration, making the approach scalable to large networks.

Our method, FIRE, orthogonalizes neural network weights after training on the current dataset but before learning on new data, using the Newton–Schulz iteration (Shown in Algorithm 1.) to efficiently approximate the above solution. Specifically, given a weight matrix $W \in \mathbb{R}^{m \times n}$, we apply the Newton–Schulz update defined as $X_{k+1} = aX_k + bX_k(X_k^{\top}X_k)$ with constants $a = 1.5$ and $b = -0.5$, where $X_0 = W/\|W\|_F$. This iterative process progressively drives the singular values of $W$ toward 1, thereby enforcing approximate orthonormality.

**Algorithm 1:** Newton–Schulz Iteration (Pytorch-like)

```
# X: a two-dimensional matrix
# N: number of iteration
a, b = (1.5, -0.5)
X = X / X.norm()
for _ in range(N):
    A = X.T @ X
    X = a * X + b * (X @ A)
return X
```

In convolutional layers, the update is applied kernel-wise along the spatial dimensions, ensuring that each convolutional filter is orthogonalized independently.

## 4 EXPERIMENTS

To demonstrate the effectiveness of FIRE, we evaluated it on three settings: continual visual learning, continual pretraining of LLMs, and reinforcement learning.

### 4.1 CONTINUAL VISUAL LEARNING

In continual visual learning experiments, we evaluated FIRE on various dataset–architecture pairs: the CIFAR-10 dataset with ResNet-18, the CIFAR-100 dataset with ViT-Tiny, and the Tiny ImageNet dataset with VGG-16.

To evaluate both the ability of FIRE to recover plasticity and its capacity to restore performance after a reset, we compare against two representative reset-based baselines: S&P (Ash & Adams, 2020), and DASH (Shin et al., 2024). Also, to evaluate the impact of regularization on the convergence speed, we adopt Parseval regularization (Chung et al., 2024), which constrains the weights to remain close to orthogonal, as a baseline. We also used L2init (Kumar et al., 2025b) as a baseline, which is a representative regularization-based method. Node-resetting methods such as Continual Backprop (CBP) (Dohare et al., 2024), Self-Normalized Resets (SNR) (Farias & Jozefiak, 2024), and Recycling Dormant Neurons (ReDo) (Sokar et al., 2023) are employed as baselines. In addition, we use the Muon optimizer (Jordan et al.) as a baseline, as it also employs the Newton–Schulz iteration. While FIRE minimizes the Deviation from Isometry (DfI) when resetting, Parseval regularization enforces the same constraint continuously throughout training. Thus, both FIRE and Parseval regularization share the same underlying optimization objective. Detailed experiment settings for continual visual learning are provided in Appendix E.1.

Following Ash & Adams (2020) and Lee et al. (2024a), we evaluate our approach in the warm-start setting, where the model is first trained on a subset of the dataset and then on the full dataset. Since plasticity loss is most severe when the subset ratio is small (Lee et al., 2024a), we warm-start with only 10% of the data before continuing on the entire dataset. As shown in Figure 2 (a), FIRE provides

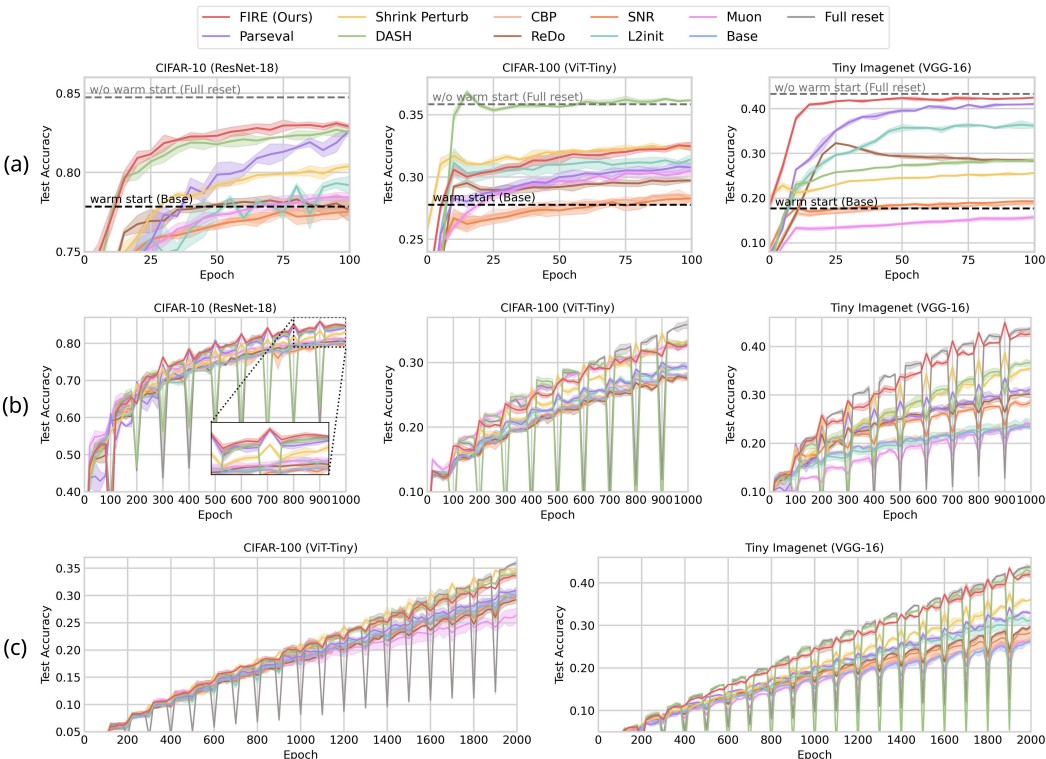

Figure 2: **Continual visual learning results.** Warm-start setting (a): training begins with only 10% of the data before continuing on the full dataset. Continual setting (b): the dataset is revealed in ten stages, expanding from 10% to 100% in 10% increments. Class-incremental setting (c): new classes are introduced over 20 phases, with an equal number of classes added at each phase.

consistent performance gains across all three benchmarks. In particular, on CIFAR-10 with ResNet-18 and Tiny ImageNet with VGG-16, FIRE outperforms all baselines, demonstrating a strong ability to recover plasticity. On CIFAR-100 with ViT-Tiny, the improvement is less pronounced. However FIRE still outperforms all other baselines except DASH, and competitive to S&P. In this setting, DASH, which employs a data-dependent shrinking strategy, proves especially effective, suggesting that guidance from data can be beneficial when reinitializing transformer architectures. Notably, Parseval regularization and L2init converges more slowly than FIRE, particularly on CIFAR-10 with ResNet-18 and Tiny ImageNet with VGG-16. This suggests that continuously enforcing the constraint during training can hinder convergence, leading to longer training and incurring additional computational cost.

To examine whether these findings hold in a setting where data are continuously added, which is a more realistic and natural setting, we evaluate FIRE in the continual setting (Lee et al., 2024a). Here, training is divided into ten stages, starting with 10% of the dataset and adding an additional 10% at each stage. In this way, data gradually expand from 10% to the full 100%. As shown in Figure 2 (b), FIRE delivers consistent gains across all datasets. The improvements are particularly pronounced on CIFAR-10 with ResNet-18 and Tiny ImageNet with VGG-16, while on CIFAR-100 with ViT-Tiny it achieves performance comparable to the best alternatives. In contrast, full reset and DASH suffer a sharp drop immediately after each reset, and although S&P avoids such drops, its performance remains suboptimal compared to FIRE. In contrast, FIRE incurs only a slight or negligible drop, suggesting that it successfully balances stability and plasticity, thereby achieving high performance with minimal drop in performance.

To assess the effectiveness of FIRE under large distribution shifts, we conducted experiments in a class-incremental learning scenario, which is widely used setup in the continual learning literature (Rebuffi et al., 2017; Dohare et al., 2024; Lewandowski et al., 2024b). New classes were gradually

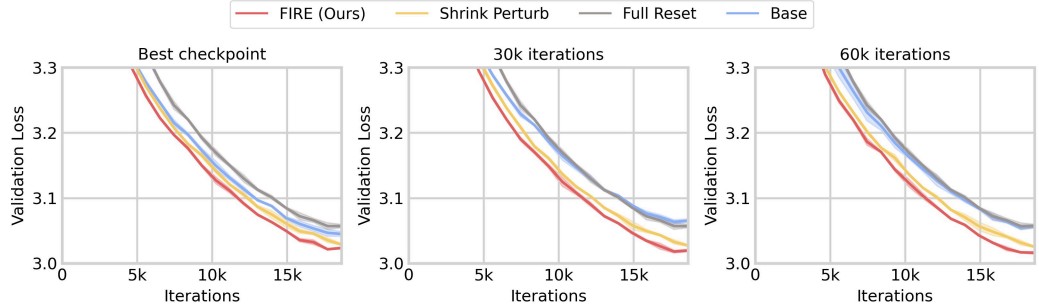

Figure 3: **Continual pretraining of GPT-0.1B.** Models are first pretrained on WikiText-103 and then continually trained on a new dataset consisting of a mixture of OpenWebText and WikiText-103. From left to right, results correspond to models initialized from the best checkpoint during pretraining, from 30k pretraining iterations, and from 60k pretraining iterations.

introduced at regular intervals. The training process was divided into 20 phases, with an equal number of classes added in each phase. Since CIFAR-10 does not contain a sufficient number of classes for this setting, we exclude it in this experiment. Figure 2 (c) reports the results in the class-incremental setting. Consistent with our earlier findings, FIRE shows strong performance without exhibiting a performance drop after resets by effectively balancing stability and plasticity, and full reset and DASH show sharp drop after reset while S&P show suboptimal performance.

Node-resetting methods such as CBP, SNR, and ReDo show poor overall performance. This is consistent with Lee et al. (2024a), which found that methods aiming to improve plasticity by maintaining trainability provide only limited gains in generalization. The Muon optimizer also performs poorly overall, suggesting that periodically reinitializing weights using the Newton–Schulz iteration (FIRE) is substantially more effective than applying this iteration to the gradients (Muon).

## 4.2 CONTINUAL PRETRAINING OF LLMS

**Setup.** We also tested FIRE in the continual pretraining of LLMs. We first pretrained a GPT-0.1B model on WikiText-103 and then trained on a combination of OpenWebText and WikiText-103. For the second phase, we used the best, 30k, and 60k checkpoints from initial pretraining to examine how plasticity loss worsens beyond the best checkpoint and how effectively FIRE mitigates this degradation at different stages. We present the detailed settings for the LLM experiments in Appendix E.2.

**Results.** As shown in Figure 3, the gap between the base model and full reset narrows as pretraining progresses, since the base model's validation loss increases with longer training. This aligns with prior findings that plasticity loss becomes more severe as pretraining duration grows (Ash & Adams, 2020). While S&P improves performance by moving parameters toward intermediate trade-off points between stability and plasticity, it remains suboptimal compared to FIRE, which achieves a more principled balance. Notably, FIRE was applied without any tuning, using a fixed 5 iterations, whereas S&P was carefully tuned over varying reinitialization degrees. Moreover, while the performance of the base model deteriorates with longer pretraining, FIRE maintains strong performance even when initialized from the 60k checkpoint. This demonstrates that FIRE can effectively balance the stability–plasticity trade-off even under severe plasticity loss.

In addition, unlike in continual visual learning (Section 4.1), full reset performs poorly in this setting. The main reason is the lack of stability inherent to full reset. Consequently, the full resetted model cannot outperform the base model, even though the base model itself already suffers from plasticity loss. In other words, the instability introduced by erasing all past information outweighs the potential benefit of restoring plasticity. These findings indicate that full parameter resetting is not an effective strategy for mitigating plasticity loss in continual pretraining of LLMs. Instead of providing a stable improvement, it wastes useful prior knowledge and leads to extreme inefficiency, making it an impractical choice in this setting.

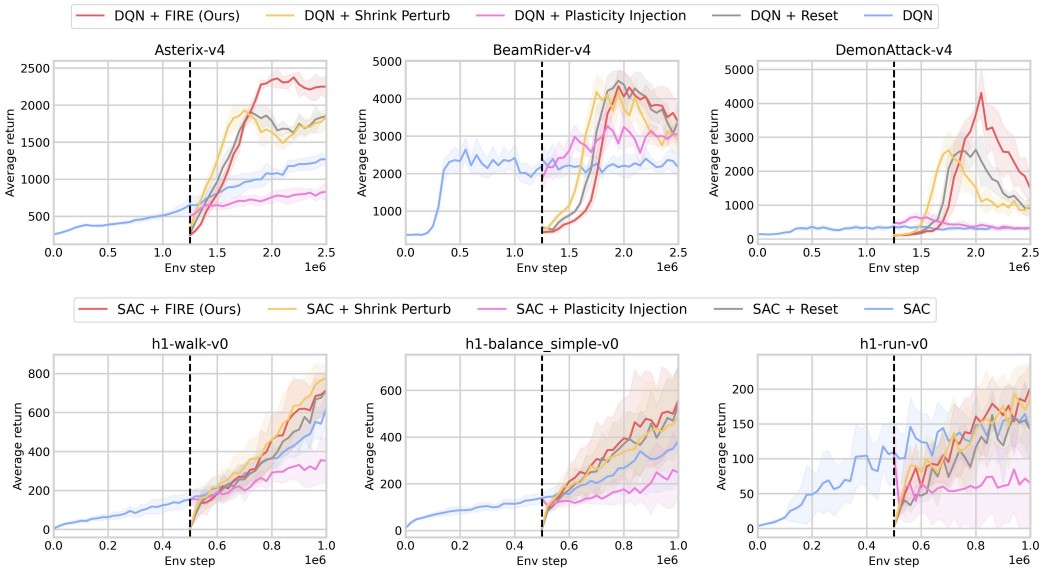

Figure 4: **Reinforcement learning results.** Discrete control with DQN on three Atari environments that suffer from severe plasticity loss (a) and continuous control with SAC on three HumanoidBench tasks (b). The black dashed line indicates the point at which reinitialization is applied.

## 4.3 REINFORCEMENT LEARNING

**Setup.** Finally, we evaluated FIRE in reinforcement learning. We evaluated the effectiveness of FIRE in a high Replay Ratio (RR) setting (Nikishin et al., 2022; Sokar et al., 2023), where loss of plasticity is severe and acts as a critical bottleneck for sample efficiency. For a comprehensive evaluation, we consider both continuous and discrete control environments. For discrete control, we focus on three Atari 2600 (Bellemare et al., 2013) games (Asterix, BeamRider, and DemonAttack), which have been reported to suffer from severe plasticity loss (Sokar et al., 2023). We use standard nature CNN with DQN algorithm (Mnih et al., 2015). For continuous control, we choose three primary tasks from HumanoidBench (Mnih et al., 2015): balance, walk, and run. We use SimBa (Lee et al., 2024b) with SAC algorithm (Haarnoja et al., 2018) as our baseline, whose replay ratio has failed to scale beyond 1 without resets (Lee et al., 2025). We considered three baselines: full reset, Shrink and Perturb (S&P) (Ash & Adams, 2020; D'Oro et al., 2022), and Plasticity Injection (Nikishin et al., 2023). To eliminate performance differences caused by randomness before reinitialization, we reinitialized the network using the same checkpoint and replay buffer.

**Results.** As shown in Figure 4, FIRE achieves superior or competitive performance across environments compared to S&P, Plasticity Injection, and full reset. In DQN, FIRE consistently outperforms S&P, surpasses full reset in Asterix, and remains competitive in other environments. Although S&P provides a slight improvement in convergence speed, it is still suboptimal relative to both full reset and FIRE. In continuous control tasks, S&P performs competitively, but it falls short of FIRE in all Atari environments. Plasticity Injection, which introduces additional parameters to balance stability and plasticity, shows poor performance across discrete and control tasks. These results suggest that manually tuning hyperparameters to balance stability and plasticity is less effective in visual reinforcement learning—where plasticity loss is particularly severe—than our principled approach, FIRE, which explicitly balances the two.

## 4.4 ABLATION STUDY

To better understand the underlying factor of FIRE's strong performance, we conducted an ablation study. To verify whether FIRE indeed effectively balances stability and plasticity, we evaluated the stability metric (SFE) and the plasticity metric (DfI), and compared FIRE against reinitialization baselines. In addition, we measured the loss landscape curvature with respect to upcoming data immediately after a reset, to examine whether our theoretical findings are also reflected in practice.

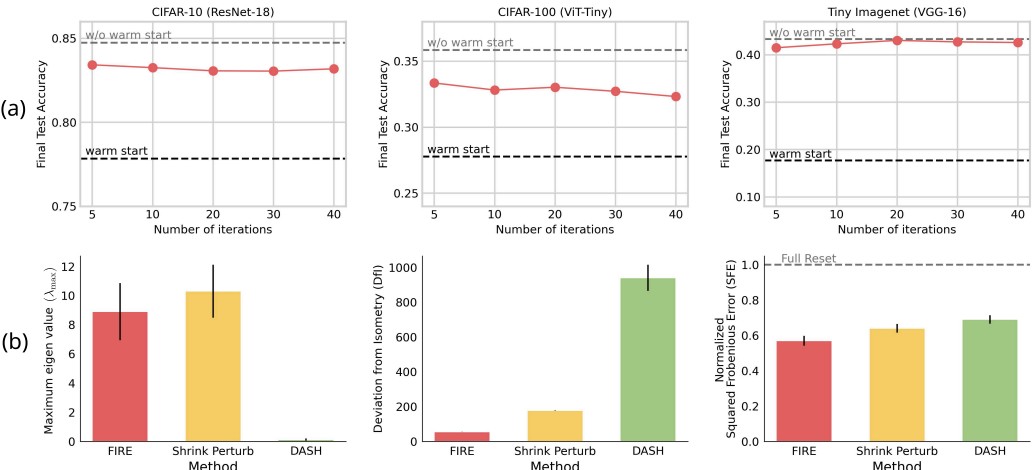

Figure 5: **Ablation study results.** Final performance of FIRE with varying numbers of iterations for Netwon-Schulz algorithm (a). Comparison of FIRE and baselines in terms of loss curvature (maximum eigenvalue of the Hessian), plasticity (DfI), and stability (normalized SFE) (b).

As shown in Figure 5 (b), FIRE achieves the lowest DfI while maintaining the lowest SFE, which suggests that FIRE successfully balances stability and plasticity in practice. Moreover, FIRE produces a smoother loss landscape compared to S&P, while still preserving a lower SFE. This indicates that our theoretical insights on DfI and loss curvature are indeed manifested in practice. Although DASH is particularly effective in smoothing the loss landscape, it also exhibits the highest SFE, which may contribute to an erasure of useful learned knowledge, thereby leading to instability after reset.

In addition, we evaluated FIRE with various hyperparameters in the warm-start setting to assess its sensitivity. The only hyperparameter in FIRE is the number of iterations for the Newton–Schulz iteration. As the number of iterations increases, we obtain a more accurate estimate of the solution to the constrained optimization problem discussed in Section 3.3. Therefore, our interest is to identify the minimum number of iterations that provides a sufficiently accurate estimate of the solution to yield performance benefits. As shown in Figure 5 (a), FIRE is highly robust to the number of iterations and already provides strong performance gains even with as few as five iterations.

## 5  CONCLUSION

In this work, we addressed stability–plasticity trade-off, which is the long-standing problem in continual learning, by introducing FIRE. By approaching stability-plasticity tradeoff as a constrained optimization problem, FIRE enables a principled reinitialization without heavy hyperparameter tuning. Across continual visual learning, reinforcement learning, and language learning benchmarks, FIRE achieved superior or competitive performance, underscoring the importance of effective stability–plasticity management for advancing continual learning.

The main limitation of our work is the assumption of access to past data. Since our focus is on balancing stability and plasticity when such access is available, we did not evaluate FIRE under restricted data scenarios. Future work should, therefore, examine FIRE under restricted access to past data. In addition, we only used relatively small models for continual pretraining of LLMs. Evaluating FIRE on the larger models and applying FIRE not only pretraining, but also continual fine-tuning of LLMs can be a promising direction for future works.

REPRODUCIBILITY STATEMENT

We provide hyperparameter configurations and implementation details of our experiments in Appendix E. The core algorithmic part of our method is described in Algorithm 1. The proofs and assumption of theoretical works provided in this paper are described in Appendix A.

ACKNOWLEDGMENTS

This work was supported by the National Research Foundation of Korea (NRF) grant funded by the Korea government (MSIT) (RS-2025-16902996). This research was financially supported by the Ministry of Trade, Industry and Energy (MOTIE) and Korea Institute for Advancement of Technology (KIAT) through the "International Cooperative R&D program" (Grant No. P0028435). This work was supported by Institute of Information & communications Technology Planning & Evaluation (IITP) grant funded by the Korea government (MSIT) (No.2019-0-01842, Artificial Intelligence Graduate School Program (GIST)). We appreciate the high-performance GPU computing support of HPC-AI Open Infrastructure via GIST SCENT.

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

## A    PROOF OF THEOREMS

**Proof of Theorem 1**

*Proof.* Unless explicitly subscripted, $\|\cdot\|$ denotes the Frobenius norm $\|\cdot\|_F$, and $\|\cdot\|_2$ denotes the spectral (operator) norm. We also recall

$$\mathrm{SFE}(\Theta, \widetilde{\Theta}) \;:=\; \sum_{j=1}^{L} \|W^j - \widetilde{W}^j\|_F^2.$$

We will use two elementary facts.

For any $F, \widetilde{F}$ with $m := \min\{\|F\|, \|\widetilde{F}\|\} > 0$,

$$\left\| \frac{F}{\|F\|} - \frac{\widetilde{F}}{\|\widetilde{F}\|} \right\| \;\leq\; \frac{2}{m}\, \|F - \widetilde{F}\|.$$

If arbitrary two matrix $U$ and $V$ satisfies $\|U\| = \|V\| = 1$ then

$$\|UU^\top - VV^\top\| \;\leq\; 2\,\|U - V\|.$$

Combining these two facts with $U := F/\|F\|$ and $V := \widetilde{F}/\|\widetilde{F}\|$ yields

$$\left\| \frac{FF^\top}{\|F\|^2} - \frac{\widetilde{F}\,\widetilde{F}^\top}{\|\widetilde{F}\|^2} \right\| \;\leq\; \frac{4}{m}\, \|F - \widetilde{F}\|. \tag{6}$$

Applying equation 6 with $F = H_\Theta^\ell(Z)$ and $\widetilde{F} = H_{\widetilde{\Theta}}^\ell(Z)$, and $m_\ell := \min\{\|H_\Theta^\ell(Z)\|, \|H_{\widetilde{\Theta}}^\ell(Z)\|\}$ gives

$$\|C_\Theta^\ell(Z) - C_{\widetilde{\Theta}}^\ell(Z)\| \;\leq\; \frac{4}{m_\ell}\, \|H_\Theta^\ell(Z) - H_{\widetilde{\Theta}}^\ell(Z)\|. \tag{7}$$

Introduce the *hybrid* outputs $H_{(\leq j)}^\ell(Z)$: the first $j$ layers use $\Theta$ and the remaining $j+1, \ldots, \ell$ layers use $\widetilde{\Theta}$. Note that $H_{(\leq \ell)}^\ell(Z) = H_\Theta^\ell(Z)$ and $H_{(\leq 0)}^\ell(Z) = H_{\widetilde{\Theta}}^\ell(Z)$. Then

$$H_\Theta^\ell(Z) - H_{\widetilde{\Theta}}^\ell(Z) = \sum_{j=1}^{\ell} \Big( H_{(\leq j)}^\ell(Z) - H_{(\leq j-1)}^\ell(Z) \Big),$$

so by the triangle inequality,

$$\|H_\Theta^\ell(Z) - H_{\widetilde{\Theta}}^\ell(Z)\| \;\leq\; \sum_{j=1}^{\ell} \|H_{(\leq j)}^\ell(Z) - H_{(\leq j-1)}^\ell(Z)\|. \tag{8}$$

Each summand differs in *only the $j$-th layer weights*. Let $X_{j-1} := H_\Theta^{j-1}(Z)$ denote the shared input fed to layer $j$ in both hybrids. Consider the *backend subnetwork*

$$\mathcal{T}_{j \to \ell}^{(\widetilde{\Theta})}(Y) := \sigma_\ell\big(\cdots \sigma_{j+1}(Y\,\widetilde{W}^{j+1}) \cdots \widetilde{W}^\ell\big).$$

For arbitrary $Y_1$ and $Y_2$, its input-Lipschitz constant is

$$\big\|\mathcal{T}_{j \to \ell}^{(\widetilde{\Theta})}(Y_1) - \mathcal{T}_{j \to \ell}^{(\widetilde{\Theta})}(Y_2)\big\| \;\leq\; \Big( \prod_{k=j+1}^{\ell} L_{\sigma_k} \|\widetilde{W}^k\|_2 \Big) \|Y_1 - Y_2\|. \tag{9}$$

Hence

$$\|H_{(\leq j)}^\ell(Z) - H_{(\leq j-1)}^\ell(Z)\| \leq \Big( \prod_{k=j+1}^{\ell} L_{\sigma_k} \|\widetilde{W}^k\|_2 \Big) L_{\sigma_j} \|X_{j-1}\| \|W^j - \widetilde{W}^j\|. \tag{10}$$

Meanwhile,

$$\|X_{j-1}\| = \|H_\Theta^{j-1}(Z)\| \leq \|Z\| \prod_{k=1}^{j-1} L_{\sigma_k} \|W^k\|_2. \tag{11}$$

Combining equation 8, equation 10, and equation 11 yields

$$\|H_\Theta^\ell(Z) - H_{\widetilde{\Theta}}^\ell(Z)\| \leq \|Z\| \Big( \prod_{k=1}^\ell L_{\sigma_k} \Big) \sum_{j=1}^\ell \Big( \prod_{k \neq j} B_k \Big) \|W^j - \widetilde{W}^j\|.$$

By Cauchy–Schwarz,

$$\sum_{j=1}^\ell \Big( \prod_{k \neq j} B_k \Big) \|W^j - \widetilde{W}^j\| \leq B_\Pi^\ell \Big( \sum_{j=1}^\ell \frac{1}{B_j^2} \Big)^{1/2} \sqrt{\mathrm{SFE}(\Theta, \widetilde{\Theta})}.$$

Therefore,

$$\|H_\Theta^\ell(Z) - H_{\widetilde{\Theta}}^\ell(Z)\| \leq \|Z\| \Big( \prod_{k=1}^\ell L_{\sigma_k} \Big) B_\Pi^\ell \Big( \sum_{j=1}^\ell \frac{1}{B_j^2} \Big)^{1/2} \sqrt{\mathrm{SFE}(\Theta, \widetilde{\Theta})}. \tag{12}$$

Substitute equation 12 into equation 7:

$$\|C_\Theta^\ell(Z) - C_{\widetilde{\Theta}}^\ell(Z)\| \leq \frac{4\|Z\|}{m_\ell} \Big( \prod_{k=1}^\ell L_{\sigma_k} \Big) B_\Pi^\ell \Big( \sum_{j=1}^\ell \frac{1}{B_j^2} \Big)^{1/2} \sqrt{\mathrm{SFE}(\Theta, \widetilde{\Theta})}.$$

Squaring both sides gives

$$\|C_\Theta^\ell - C_{\widetilde{\Theta}}^\ell\|^2 \leq \frac{16\|Z\|^2}{m_\ell^2} \Big( \prod_{k=1}^\ell L_{\sigma_k} \Big)^2 (B_\Pi^\ell)^2 \Big( \sum_{j=1}^\ell \frac{1}{B_j^2} \Big) \mathrm{SFE}(\Theta, \widetilde{\Theta}).$$

If $L_{\sigma_k} \leq 1$ and $B_j \leq S$ for all $j$, then $(B_\Pi^\ell)^2 \sum_{j=1}^\ell B_j^{-2} \leq \ell\, S^{2\ell-2}$. Therefore

$$\|C_\Theta^\ell - C_{\widetilde{\Theta}}^\ell\| \leq \frac{4\|Z\|}{m_\ell} \sqrt{\ell}\, S^{\ell-1} \sqrt{\mathrm{SFE}(\Theta, \widetilde{\Theta})}.$$

$\square$

**Network, Loss, and Notation.** Let $Z \in \mathbb{R}^{n \times d_0}$ be the input matrix and $W_k \in \mathbb{R}^{d_{k-1} \times d_k}$ ($k = 1, \ldots, L$) be the weight matrices. Define

$$A_k = H_{k-1} W_k \in \mathbb{R}^{n \times d_k}, \qquad H_k = \rho(A_k) \quad (\rho = \mathrm{ReLU}), \qquad U := A_L \in \mathbb{R}^{n \times d_L},$$

with $H_0 := Z$. The empirical risk is

$$\mathcal{L}(W_{1:L}) = \frac{1}{n} \sum_{i=1}^n \ell_i(u_i), \qquad u_i \in \mathbb{R}^{d_L}.$$

Let $\theta = \mathrm{vec}(W_1, \ldots, W_L) \in \mathbb{R}^p$ and $H_\theta := \nabla_\theta^2 \mathcal{L} \in \mathbb{R}^{p \times p}$. We use $\|\cdot\|_2$ for spectral norm and $\|\cdot\|_F$ for the Frobenius norm.

We denote the maximum eigen values as $\lambda_{\max}$

**Deviation From Isometry (DfI).** For a matrix $W$, $\mathrm{DfI}(W) := \|W^\top W - I\|_F^2$. For each layer, set

$$\nu_k := 1 + \sqrt{\mathrm{DfI}(W_k)}, \qquad \alpha_k := \sqrt{\nu_k}.$$

First, let us examine the lemmas required for the proof of Theorem 2.

**Lemma 1** (DfI controls the spectral norm). *For each layer $k$, $\|W_k\|_2^2 \leq \nu_k$ and $\|W_k\|_2 \leq \alpha_k$.*

*Proof.* $\|W\|_2^2 = \lambda_{\max}(W^\top W) \leq \lambda_{\max}(W^\top W - I) + 1 \leq \|W^\top W - I\|_2 + 1 \leq \|W^\top W - I\|_F + 1 = 1 + \sqrt{\mathrm{DfI}(W)}$. $\qquad \square$

**Lemma 2** (Covariance/spectral growth through layers). *Let $\Sigma_{H_k} = \frac{1}{n} H_k^\top H_k$. Then*

$$\lambda_{\max}(\Sigma_{H_k}) \leq \lambda_{\max}(\Sigma_{H_{k-1}}) \|W_k\|_2^2 \leq \lambda_{\max}(\Sigma_{H_{k-1}}) \nu_k.$$

*Consequently, with $\Sigma_{H_0} = \Sigma_Z$,*

$$\lambda_{\max}(\Sigma_{H_k}) \leq \lambda_{\max}(\Sigma_Z) \prod_{j=1}^{k} \nu_j.$$

*In particular, under $\Sigma_Z \approx I$ we have $\lambda_{\max}(\Sigma_{H_k}) \leq \prod_{j=1}^{k} \nu_j$.*

*Proof.* ReLU is 1-Lipschitz (applied elementwise), hence $\|H_k\|_2 \leq \|A_k\|_2 \leq \|H_{k-1}\|_2 \|W_k\|_2$. Therefore $\frac{1}{n}\|H_k\|_2^2 \leq \frac{1}{n}\|H_{k-1}\|_2^2 \|W_k\|_2^2$, i.e., $\lambda_{\max}(\Sigma_{H_k}) \leq \lambda_{\max}(\Sigma_{H_{k-1}})\|W_k\|_2^2$. Apply Lemma 1. $\qquad \square$

**Lemma 3** (Block-Jacobian bound). *Let $J \in \mathbb{R}^{(nd_L) \times p}$ be the Jacobian of $\mathrm{vec}(U)$ w.r.t. $\theta$, and $J = [J_1\ J_2\ \cdots\ J_L]$ the block-columns corresponding to $\mathrm{vec}(W_k)$. Then*

$$\frac{1}{n}\|J_k\|_2^2 \leq \Big(\prod_{j=1}^{k-1} \nu_j\Big) \Big(\prod_{j=k+1}^{L} \nu_j\Big) = \prod_{j \neq k} \nu_j.$$

*Consequently,*

$$\frac{1}{n}\|J\|_2^2 \leq \sum_{k=1}^{L} \frac{1}{n}\|J_k\|_2^2 \leq \sum_{k=1}^{L} \prod_{j \neq k} \nu_j.$$

*Proof.* Consider a perturbation $\Delta W_k$. Without loss of generality, we may assume $\|\Delta W_k\|_F = 1$, since the operator norm is defined by the supremum over unit perturbations. With fixed ReLU gates (op. norm $\leq 1$), the output perturbation over all $n$ samples satisfies

$$\Delta U = H_{k-1} \Delta W_k B_{k+1:L}, \quad B_{k+1:L} := \underbrace{D_k W_{k+1} D_{k+1} \cdots D_{L-1}}_{\text{diag gates, } \|\cdot\|_2 \leq 1} W_L.$$

Thus $\|\Delta U\|_F \leq \|H_{k-1}\|_2 \|\Delta W_k\|_F \|B_{k+1:L}\|_2$, so the operator norm of the linear map $\Delta W_k \mapsto \Delta U$ is at most $\|H_{k-1}\|_2 \|B_{k+1:L}\|_2$. Hence $\|J_k\|_2 \leq \|H_{k-1}\|_2 \|B_{k+1:L}\|_2$ and

$$\frac{1}{n}\|J_k\|_2^2 \leq \frac{1}{n}\|H_{k-1}\|_2^2 \|B_{k+1:L}\|_2^2 = \lambda_{\max}(\Sigma_{H_{k-1}}) \|B_{k+1:L}\|_2^2.$$

Using Lemma 2, $\lambda_{\max}(\Sigma_{H_{k-1}}) \leq \prod_{j=1}^{k-1} \nu_j$. Also $\|B_{k+1:L}\|_2 \leq \prod_{j=k+1}^{L} \|W_j\|_2 \leq \prod_{j=k+1}^{L} \alpha_j$, thus $\|B_{k+1:L}\|_2^2 \leq \prod_{j=k+1}^{L} \nu_j$. Multiplying the two bounds yields the claim. Finally, since $J$ is a horizontal concatenation of blocks, $\|J\|_2^2 \leq \sum_k \|J_k\|_2^2$. $\qquad \square$

**Lemma 4** (Gauss–Newton part). *For each sample, $\nabla_\theta^2 \ell_i = J_i^\top (\nabla_u^2 \ell_i) J_i + R_i$ with some remainder $R_i$. By the second assumption, $\|\nabla_u^2 \ell_i\|_2 \leq \beta$, hence*

$$\left\| \frac{1}{n} \sum_{i=1}^{n} J_i^\top (\nabla_u^2 \ell_i) J_i \right\|_2 \leq \frac{\beta}{n} \|J\|_2^2 \leq \beta \sum_{k=1}^{L} \prod_{j \neq k} \nu_j,$$

*where the last inequality uses Lemma 3.*

**Lemma 5** (Remainder term). *Let $R := \frac{1}{n} \sum_{i=1}^{n} R_i$. Under the second and third assumption,*

$$\|R\|_2 \leq 2\gamma \sum_{1 \leq k < \ell \leq L} \prod_{j \notin \{k,\ell\}} \nu_j.$$

*Proof.* Fix the ReLU gates locally (piecewise linear region) and $\sum_k \|\Delta W_k\|_F = 1$, since the operator norm is defined by the supremum over unit perturbations. Then the network output $U$ is *multilinear* in $\{W_k\}_{k=1}^{L}$. For $k < \ell$, the mixed second derivative block maps $(\Delta W_k, \Delta W_\ell)$ to

$$H_{k-1} \, \Delta W_k \, C_{k+1:\ell-1} \, \Delta W_\ell \, B_{\ell+1:L},$$

where $C_{k+1:\ell-1}$ is the product of intermediate gated weights, and $B_{\ell+1:L}$ the tail product as in Lemma 3. By submultiplicativity,

$$\|H_{k-1}\Delta W_k C_{k+1:\ell-1}\Delta W_\ell B_{\ell+1:L}\|_F \leq \|H_{k-1}\|_2 \|\Delta W_k\|_F \|C_{k+1:\ell-1}\|_2 \|\Delta W_\ell\|_F \|B_{\ell+1:L}\|_2.$$

Using Lemma 2 and Lemma 1,

$$\|H_{k-1}\|_2 \leq \sqrt{n} \prod_{j=1}^{k-1} \alpha_j, \quad \|C_{k+1:\ell-1}\|_2 \leq \prod_{j=k+1}^{\ell-1} \alpha_j, \quad \|B_{\ell+1:L}\|_2 \leq \prod_{j=\ell+1}^{L} \alpha_j.$$

Therefore, after dividing by $n$ (from the prefactor $1/n$ in $\mathcal{L}$) and summing the symmetric contribution $(\ell, k)$, the bilinear remainder contributes at most

$$\frac{2}{n} \sum_{k<\ell} \|H_{k-1}\|_2 \|C_{k+1:\ell-1}\|_2 \|B_{\ell+1:L}\|_2 \|\Delta W_k\|_F \|\Delta W_\ell\|_F \leq 2 \sum_{k<\ell} \left( \prod_{j \notin \{k,\ell\}} \alpha_j^2 \right) \|\Delta W_k\|_F \|\Delta W_\ell\|_F.$$

Finally, by $2ab \leq a^2 + b^2$ and $\sum_k \|\Delta W_k\|_F^2 = 1$ (unit parameter direction), the operator norm of the second-derivative map is bounded by $\sum_{k<\ell} \prod_{j \notin \{k,\ell\}} \nu_j$. Multiplying by $\|\nabla_u \ell_i\|_2 \leq \gamma$ and averaging over $i$ gives the claim. $\square$

**Proof of Theorem 2**

*Proof.* Combine Lemma 4 and Lemma 5 and use $\|A + B\|_2 \leq \|A\|_2 + \|B\|_2$. $\square$

**Corollary 1** (Near-interpolation or small-gradient regime). *If the training gradients at the outputs are small so that $\gamma \approx 0$, then*

$$\left\|\nabla_\theta^2 \mathcal{L}(W_{1:L})\right\|_2 \lesssim \beta \sum_{k=1}^{L} \prod_{j \neq k} \left(1 + \sqrt{\mathrm{DfI}(W_j)}\right).$$

Next, we present the lemma required for the proof of Theorem 3.

**Lemma 6** (A basic spectral lemma from DfI). *Let $\varepsilon = \sqrt{\mathrm{DfI}(W)}$. Then*

$$\|W^\top W - I\|_2 \leq \|W^\top W - I\|_F = \varepsilon,$$

*hence every eigenvalue $\mu$ of $W^\top W = S^2$ satisfies $1 - \varepsilon \leq \mu \leq 1 + \varepsilon$. Equivalently,*

$$\sqrt{1 - \varepsilon}\, I \preceq S \preceq \sqrt{1 + \varepsilon}\, I.$$

*Proof.* By the definition of $\varepsilon$, we have

$$\|W^\top W - I\|_2 \leq \|W^\top W - I\|_F = \varepsilon.$$

Therefore, all eigenvalues $\mu$ of $W^\top W$ lie within the interval

$$1 - \varepsilon \leq \mu \leq 1 + \varepsilon.$$

Since $W^\top W = S^2$ with $S \succeq 0$, this is equivalent to the spectral bound

$$\sqrt{1 - \varepsilon}\, I \preceq S \preceq \sqrt{1 + \varepsilon}\, I.$$

$\square$

Using this Lemma, we provide proof of Theorem 3 below:

**Proof of Theorem 3**

*Proof.* Let $Z \in \mathbb{R}^{n \times a}$ be the input matrix and $W \in \mathbb{R}^{a \times b}$ a weight matrix. The resulting feature matrix is $\Phi = ZW \in \mathbb{R}^{n \times b}$ and the empirical covariances are

$$\Sigma_Z = \tfrac{1}{n} Z^\top Z \in \mathbb{R}^{a \times a}, \qquad \Sigma_\Phi = \tfrac{1}{n} \Phi^\top \Phi = W^\top \Sigma_Z W \in \mathbb{R}^{b \times b}.$$

Let $W = QS$ denote the right polar decomposition of $W$, where $Q \in \mathbb{R}^{a \times b}$ has orthonormal columns $(Q^\top Q = I_b)$ and $S = (W^\top W)^{1/2} \in \mathbb{R}^{b \times b}$ is positive definite. Then

$$\Sigma_\Phi = W^\top \Sigma_Z W = S(Q^\top \Sigma_Z Q) S.$$

Write $M := Q^\top \Sigma_Z Q \succeq 0$, and let its positive eigenvalues be $\eta_1 \geq \cdots \geq \eta_d > 0$, where $d = \operatorname{rank}(M) \leq \min\{b, \operatorname{rank}(\Sigma_Z)\}$. Let $\sigma_1(\Phi) \geq \cdots \geq \sigma_d(\Phi) > 0$ denote the nonzero singular values of $\Phi$.

For any $x \in \mathbb{R}^b$ with $\|x\| = 1$,

$$x^\top \Sigma_\Phi x = x^\top S M S x = (Sx)^\top M (Sx).$$

Let $y = Sx$. Then $x^\top \Sigma_\Phi x = y^\top M y$ and, by Lemma 6,

$$\|y\|_2^2 = \|Sx\|_2^2 \in [1 - \varepsilon, 1 + \varepsilon].$$

Therefore

$$(1 - \varepsilon) \lambda_{\min}^+(M) \leq x^\top \Sigma_\Phi x \leq (1 + \varepsilon) \lambda_{\max}(M),$$

where $\lambda_{\min}^+(M) = \eta_d$ denotes the smallest positive eigenvalue of $M$ (the lower bound is interpreted on the subspace where $My \neq 0$). Taking the maximum over unit $x$ yields

$$\lambda_{\max}(\Sigma_\Phi) \leq (1 + \varepsilon) \eta_1,$$

and taking the minimum Rayleigh quotient over the orthogonal complement of $\ker(\Sigma_\Phi)$ yields

$$\lambda_{\min}^+(\Sigma_\Phi) \geq (1 - \varepsilon) \eta_d.$$

Since $\sigma_{\max}(\Phi)^2 = n \lambda_{\max}(\Sigma_\Phi)$ and $\left(\sigma_{\min}^+(\Phi)\right)^2 = n \lambda_{\min}^+(\Sigma_\Phi)$, below inequality holds with $d = \operatorname{rank}(M)$.

$$\sqrt{n} \sqrt{1 - \varepsilon} \sqrt{\eta_d} \leq \sigma_{\min}^+(\Phi) \leq \sigma_{\max}(\Phi) \leq \sqrt{n} \sqrt{1 + \varepsilon} \sqrt{\eta_1}.$$

Here $\sigma_{\min}^+(\Phi)$ denotes the smallest positive singular value of $\Phi$ (defined only when $d \geq 1$).

Therefore, if $d \geq 1$, then

$$\rho_\Phi := \frac{\sigma_{\max}(\Phi)}{\sigma_{\min}^+(\Phi)} \leq \sqrt{\frac{1 + \varepsilon}{1 - \varepsilon}} \cdot \sqrt{\frac{\eta_1}{\eta_d}}. \tag{13}$$

Consider the worst-case allocation of the nonzero singular values that maximizes the cumulative ratio $\sum_{i=1}^k \sigma_i / \sum_{i=1}^d \sigma_i$ given a fixed condition number bound $\rho_\Phi$: the top $k$ singular values all equal $\sigma_{\max}$ and the remaining $d - k$ equal $\sigma_{\min}^+$. Then

$$\frac{\sum_{i=1}^k \sigma_i(\Phi)}{\sum_{i=1}^d \sigma_i(\Phi)} \leq \frac{k \sigma_{\max}}{k \sigma_{\max} + (d - k) \sigma_{\min}^+} = \frac{k \rho_\Phi}{k \rho_\Phi + (d - k)}. \tag{14}$$

To achieve a coverage level of $1 - \delta$ with $k$ singular values, it is necessary that

$$\frac{k \rho_\Phi}{k \rho_\Phi + (d - k)} \geq 1 - \delta \implies k \geq \frac{(1 - \delta) d}{\delta \rho_\Phi + (1 - \delta)}.$$

Taking the ceiling and substituting the bound on $\rho_\Phi$ from (13), establishes the left inequality in (3)

When $\Sigma_Z = I$, we have $M = Q^\top I Q = I$, so $\eta_1 = \cdots = \eta_b = 1$ and $d = b$. Then,

$$\sqrt{n} \sqrt{1 - \varepsilon} \leq \sigma_i(\Phi) \leq \sqrt{n} \sqrt{1 + \varepsilon} \qquad (\forall i),$$

Which leads to

$$\rho_\Phi \leq \sqrt{\tfrac{1+\varepsilon}{1-\varepsilon}}$$

Substituting $\rho_\Phi$ in first inequality of (3) leads to the right inequality (3). Without whitening, the achievable flattening is limited by the compressed input spectrum $M = Q^\top \Sigma_Z Q$.

$\square$

Next, we present the proof for Theorem 4.

**Proof of Theorem 4**

*Proof.* By isotropy and positive homogeneity, there exists a constant $c_\sigma > 0$ such that $\mathbb{E}_z[|\sigma(\langle z, w_j \rangle)|] = c_\sigma \|w_j\|$ for all $j$. Hence $s_j = \|w_j\|/(\frac{1}{b}\sum_k \|w_k\|)$. Let $u_j = \|w_j\|^2 = [W^\top W]_{jj}$. Since

$$\mathrm{DfI}(W) = \|W^\top W - I\|_F^2 = \sum_{j=1}^{b}(u_j - 1)^2 + 2\sum_{i<j}\langle w_i, w_j\rangle^2 \geq \sum_{j=1}^{b}(u_j - 1)^2,$$

we obtain $|u_j - 1| \leq \sqrt{\sum_k (u_k - 1)^2} \leq \epsilon$, i.e., $1 - \epsilon \leq \|w_j\|^2 \leq 1 + \epsilon$ for all $j$. The same bounds imply $\sqrt{1-\epsilon} \leq \bar{r} := \frac{1}{b}\sum_k \|w_k\| \leq \sqrt{1+\epsilon}$. Therefore

$$\frac{\sqrt{1-\epsilon}}{\sqrt{1+\epsilon}} \leq s_j = \frac{\|w_j\|}{\bar{r}} \leq \frac{\sqrt{1+\epsilon}}{\sqrt{1-\epsilon}}.$$

$\square$

**Corollary 2** (Absence of $\tau$-dormant neurons)**.** *Fix $\tau \in (0, 1)$. If $\mathrm{DfI}(W) \leq \left(\frac{1-\tau^2}{1+\tau^2}\right)^2$, then $s_j \geq \tau$ for all $j$.*

Note that Sokar et al. (2023) measured neurons with a dormancy score of 0.025 or lower as dormant. In this threshold, based on our theoretical analysis, $\mathrm{DfI}(W) < 0.9975$ can eliminate dormant neurons from the network.

**Derivation of Equation 5**

Here we provide derivation of Equation 5.

First expand the norm:

$$\|W - \widetilde{W}\|_F^2 = \|W\|_F^2 + \|\widetilde{W}\|_F^2 - 2\,\mathrm{tr}(\widetilde{W}^\top W).$$

From the constraint $\widetilde{W}^\top \widetilde{W} = I$, we have $\|\widetilde{W}\|_F^2 = \mathrm{tr}(I) = n$, so

$$\min_{\widetilde{W}^\top \widetilde{W}=I} \|W - \widetilde{W}\|_F^2 \quad \Longleftrightarrow \quad \max_{\widetilde{W}^\top \widetilde{W}=I} \mathrm{tr}(\widetilde{W}^\top W).$$

Let $S := W^\top W \succ 0$, and define $Q := WS^{-1/2}$. Then

$$Q^\top Q = S^{-1/2}W^\top WS^{-1/2} = S^{-1/2}SS^{-1/2} = I,$$

so $Q$ is feasible, and

$$W = QS^{1/2}$$

is the (column) polar decomposition of $W$.

Now take any feasible $\widetilde{W}$ and set

$$Z := \widetilde{W}^\top Q.$$

Then

$$\mathrm{tr}(\widetilde{W}^\top W) = \mathrm{tr}(\widetilde{W}^\top QS^{1/2}) = \mathrm{tr}(ZS^{1/2}).$$

Because $\widetilde{W}$ and $Q$ have orthonormal columns, one can show $Z^\top Z \preceq I$, so all singular values $\sigma_i(Z)$ satisfy $0 \leq \sigma_i(Z) \leq 1$. By von Neumann's trace inequality,

$$\mathrm{tr}(ZS^{1/2}) \leq \sum_{i=1}^{n} \sigma_i(Z)\,\sigma_i(S^{1/2}) \leq \sum_{i=1}^{n} \sigma_i(S^{1/2}) = \mathrm{tr}(Q^\top W),$$

with equality when $Z = I$, i.e., when $\widetilde{W} = Q$.

Therefore the solution is:

$$\widetilde{W}^\star = Q = W(W^\top W)^{-\frac{1}{2}}.$$

# B  ADDITIONAL RESULTS

## B.1  COMPUTATIONAL EFFICIENCY

To prove computational efficiency of FIRE, we provide wall-clock time and GPU memory usage in Table 1.

Table 1: Wall-Clock Time and GPU memory footprint of FIRE and baseline methods

| Method | Wall-Clock Time | GPU Memory |
|---|---|---|
| Shrink Perturb | 0.002 sec | 27 MB |
| FIRE | 0.06 sec | 55 MB |
| DASH | 69 sec | 2834 MB |

As shown in the table, FIRE introduces negligible computational cost and memory usage similar to Shrink Perturb, while significantly efficient compard to DASH.

The result is averaged across 10 trials, on VGG16 architecture with TinyImageNet dataset. We used a machine consists of TITAN RTX 24GB GPU and AMD Ryzen 7 5800X 8-Core Processor, with 64GB RAM.

## B.2  NUMBER OF ITERATIONS FOR NEWTON-SCHULZ ITERATION

In this section, we provide a more detailed analysis which illustrates how SFE and DfI evolve across FIRE iterations.

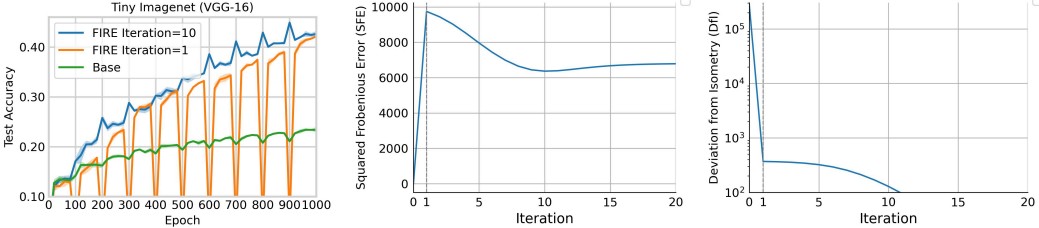

Figure 6: **Effect of number of FIRE iterations.** Test accuracy of FIRE with single iteration and 10 iterations (left). Change of SFE during FIRE iterations (middle). Change of DfI during FIRE iterations (right).

As shown in Figure 6 (right), DfI decreases substantially after only a single iteration. This suggests that using a small number of iterations ($< 5$) is sufficient to bring performance benefits. However, as shown in Figure 6 (middle), SFE reaches its peak at the first iteration and then decreases as the number of iterations increases, indicating that using only a few iterations ($< 5$) can introduce instability and ultimately lead to performance degradation.

We validate this result in continual visual learning (VGG-16 with Tiny-ImageNet). Figure 6 (left) shows comparison between FIRE with 10 iterations and with single iteration. The result shows that even with single iteration still can achieve comparable performance with 10 iterations, but show significant drop after reinitalization, which supports aforementioned findings.

## B.3  COEFFICIENTS FOR NEWTON-SCHULZ ITERATION

Our orthogonalization strategy builds on the Newton–Schulz (NS) iteration, which has also been adopted in recent works such as Muon. However, the exact recurrence used in Muon differs from ours. Muon employs a tuned quintic polynomial of the form

$$\varphi(x) = ax + bx^3 + cx^5,$$

with optimized coefficients such as $(a, b, c) = (3.4445, -4.7750, 2.0315)$, chosen to accelerate convergence so that only a few iterations are needed in practice. However, this sacrifices accuracy for speed, since the singular values do not converge to 1, but oscillate near it. Since our interest is accuracy rather than speed, we adopt the standard coefficients $(a, b, c) = (2, -1.5, 0.5)$, which correspond to a well-known rectangular variant of NS:

$$X_{k+1} = 2X_k - 1.5X_k(X_k^\top X_k) + 0.5X_k(X_k^\top X_k)^2.$$

Although this more slowly increases small singular values than Muon's tuned version, it accurately converges to orthogonal matrix.

Empirically, we did not observed significant difference in performance when we tested both coefficients in the warm-start setting (results are shown in Figure 7).

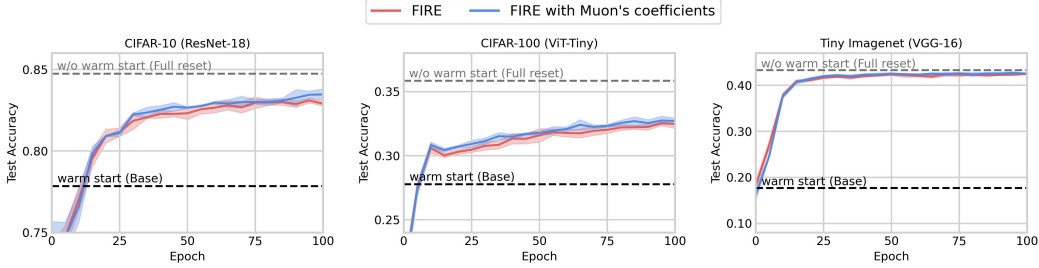

Figure 7: **Effect of Newton Schulz iteration coefficients on FIRE.** FIRE and FIRE with Muon's coefficients are evaluated on warm-start setting under CIFAR-10 with ResNet-18 (left), CIFAR-100 with ViT-Tiny (middle), and Tiny ImageNet with VGG-16 (right).

### B.4 TRAIN ACCURACY

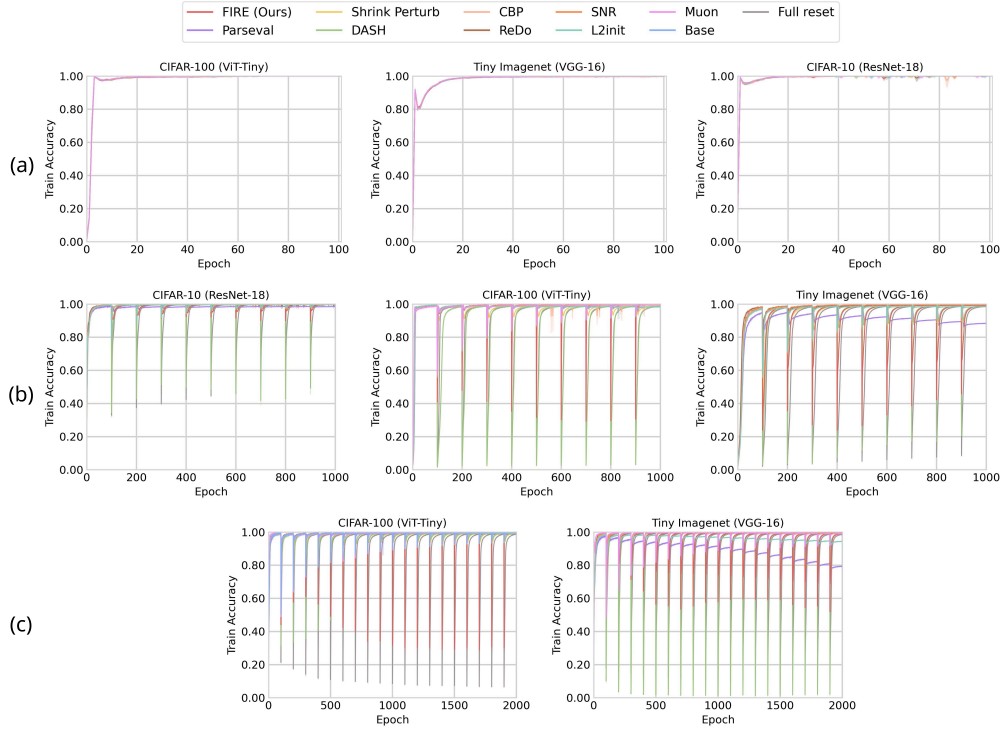

Figure 8: **Train accuracy of continual visual learning experiment.** Warm-start setting (a), Continual setting (b), and Class-incremental setting (c).

## C  IMPLEMENTATION DETAILS OF FIRE

Here we describe how FIRE is applied in practice to different modules of the network.

**Linear layers.**  For fully-connected weights $W \in \mathbb{R}^{d_{\text{out}} \times d_{\text{in}}}$, we first normalize and then apply NS iteration to approximate an orthogonal matrix. Since orthogonalization alone changes the scale of the outputs, we multiply the result by

$$\text{scale} = \sqrt{\frac{d_{\text{out}}}{d_{\text{in}}}}.$$

This factor is motivated by the Modular Duality framework (Bernstein & Newhouse, 2025), which shows that taking the ratio of output to input dimension is sufficient to preserve stable signal variance. In short, the orthogonalization ensures the weights are well-conditioned, and the scaling factor restores the right magnitude.

**Convolutional layers.**  For convolutional filters $W \in \mathbb{R}^{C_{\text{out}} \times C_{\text{in}} \times k_h \times k_w}$, we apply the same procedure slice by slice over the spatial indices. Here the scaling factor additionally accounts for the size of the kernel:

$$\text{scale} = \frac{\sqrt{C_{\text{out}}/C_{\text{in}}}}{k_h k_w}.$$

Intuitively, the larger the kernel, the more input values contribute to each output, so we divide by the kernel area to prevent the output variance from exploding.

Note that for each spatial location $(i, j)$, the slice $W[:, :, i, j] \in \mathbb{R}^{C_{\text{out}} \times C_{\text{in}}}$ is orthogonalized independently by applying the Newton–Schulz iteration.

**Attention modules.**  In Vision Transformers (ViTs), we restrict orthogonalization to the query ($Q$) and key ($K$) projections. Empirically, applying it to the feedforward MLP layers or the output projections does not provide clear benefits and may even reduce performance. Because the dot-product $QK^{\top}$ is the part most sensitive to poor conditioning, orthogonalizing $Q$ and $K$ helps improve the stability of similarity scores while leaving the value ($V$), output ($O$), and MLP weights unchanged.

# D  BASELINE METHODS

**Shrink & Perturb.** Shrink & Perturb (S&P) is a Reset-based method that shrinks weight parameters and injects noise (Ash & Adams, 2020). This method has proven particularly beneficial for warm-start training. Following the setup in prior work Lee et al. (2024a), we control both the noise level and the shrinkage strength using a single hyperparameter. Formally, letting $\theta$ denote the learnable parameters, $\theta_0$ the initial parameters, and $\lambda$ the S&P coefficient, the update rule is: $\theta \leftarrow (1 - \lambda)\theta + \lambda\theta_0$.

**DASH.** Direction-Aware SHrinking (DASH) (Shin et al., 2024) is a Reinitialization-based method that selectively shrinks network weights according to their directional alignment with the loss gradient, measured by cosine similarity. This method suppresses parameters that contribute to noise memorization while preserving weights that encode task-relevant features. This method enhances training efficiency and preserves model plasticity, leading to improved generalization under stationary data distributions. In our experiments, we applied DASH when new data was added.

**Parseval Regularization.** Parseval Reg. introduces a regularization term that enforces the weight matrices of neural network layers to remain approximately orthogonal (Chung et al., 2024). Formally, letting $W$ denote a weight matrix, $I$ identity matrix, and $\|\cdot\|_F$ the Frobenius norm. The loss term is $\lambda\|WW^\top - sI\|_F^2$, where $s > 0$ is a scaling factor and $\lambda$ is the regularization strength. It penalizes the deviation of $WW^\top$ from a scaled identity matrix, encouraging the rows of each weight matrix $W$ to be orthogonal and have controlled norms. This constraint keeps the singular values of $W$ close to a constant, preventing gradient explosion or vanishing and leading to more stable and efficient optimization. We used $s = 1$ in all experiments and only swept $\lambda$.

**Continual Backpropagation.** Continual Backpropagation (CBP) selectively reinitializes low-utility hidden units using a contribution-utility measure (Dohare et al., 2024). Contribution-utility scores are computed as an exponential moving average of the unit's activation magnitude multiplied by the summed magnitude of its outgoing weights. Units with persistently low contribution utility are considered uninformative and are periodically reset. CBP is controlled by two hyperparameters: the maturity threshold $m$, which protects units from reinitialization for at least $m$ update steps to allow stable utility estimation, and the replacement rate $\rho$, which determines the expected fraction of units to reset at each update step via fractional accumulation.

**Recycling Dormant neurons.** Recycling Dormant neurons (ReDo) (Sokar et al., 2023) is another unit-reinitialization method that assigns a neuron score to each hidden unit in every layer, and resets units whose scores fall below the hyperparameter $\tau$. The neuron score $s$ is computed as the ratio between a unit's average activation magnitude and the average activation magnitude of all units in the same layer, formally defined as $s_i^\ell = \frac{\mathbb{E}_{x \in D}|h_i^\ell(x)|}{\frac{1}{H^\ell}\sum_{k \in h}\mathbb{E}_{x \in D}|h_k^\ell(x)|}$, where $h_i^\ell(x)$ denotes the activation of neuron $i$ in layer $\ell$ for input $x \in D$, and $H^\ell$ is the number of neurons in layer $\ell$.

**L2 Init.** L2 Init (Kumar et al., 2025b), as known as Regen (Regenerative regularization), is a weight regularization method designed to mitigate plasticity loss by leveraging the property that the initial network exhibits the highest plasticity. L2 Init regularizes the weights to stay close to the initial weights by adding a term $\lambda\|W - W_0\|_F^2$ to the loss function, where $\lambda$ is the regularization strength, $W$ is the current weight matrix and $W_0$ is initial weight matrix.

**Self-Normalized Resets.** Self-Normalized Resets (SNR) (Farias & Jozefiak, 2024) is a reset-based method that detects inactive neurons by monitoring their firing statistics and statistically testing whether a neuron's activity is consistent with its past behavior. For each neuron, SNR maintains an empirical distribution of inter-firing times (the number of consecutive updates with zero activation). If the computed probability falls below a threshold $1 - \tau$, the neuron is classified as inactive and reset. This procedure adaptively replaces neurons whose activity has effectively vanished, mitigating plasticity loss without relying on a fixed, hand-tuned inactivity window.

**Muon.** Muon (Jordan et al.) is an optimizer that augments SGD with momentum by orthogonalizing its update matrices. Concretely, Muon first forms the usual SGD-momentum update $G$ for each weight matrix and then applies a small fixed number of Newton–Schulz iterations to approximate the closest semi-orthogonal matrix $Ortho(G)$, effectively replacing $G$ by a matrix with singular values near one while staying close in Frobenius norm. Following the reference implementation, in our experiments we apply Muon only to the middle weight matrices of hidden layers, while scalar

and vector parameters, as well as input and output layers, are optimized with AdamW. We set the momentum to 0.95 as recommended by the authors.

**Plasticity Injection.** Plasticity injection (Nikishin et al., 2023) restores neural network's plasticity by adding a fresh, randomly initialized copy of the prediction head while leaving current predictions unchanged at the moment of the change. The original prediction head is frozen, and two identical new heads are created, one that is allowed to learn and one that always stays fixed. At the start, the learned and fixed new heads cancel each other out, so the overall output of the agent stays exactly the same. As training continues, the learnable new head adapts to new data, giving the agent renewed flexibility, while the original and the fixed new head act as a stable reference. For DQN, we applied plasticity injection to MLP layers, and for SAC, we applied it to whole critic network which is known to suffer from severe plasticity loss (Ma et al., 2023).

# E DETAILED EXPERIMENT SETTINGS

## E.1 CONTINUAL VISUAL LEARNING

For continual visual learning, we report the results with 3 seeds.

Table 2: Detailed settings in continual visual learning.

| Parameter | Value |
|---|---|
| Optimizer | Adam (Kingma & Ba, 2014) |
| Learning Rate | 1e−3 |
| Learning Rate Scheduler | Warmup |
| Gradient norm clipping | 0.5 |
| Batch Size | 256 |
| Epochs per Chunk | 100 |
| Data Augmentation | False |

In this section, we describe the detailed settings for conducting continual visual learning. We note that most of the hyperparameters we used were adopted from Lee et al. (2024a).

For the Warmup scheduler, the learning rate is increased linearly from 0 to the target learning rate during the first 10% of training on each dataset. In other words, in the case of Table 2, the learning rate is gradually raised from 0 to 1e−3 over the first 10 epochs of each data chunk.

In the warm-start scenario described in Section 4.1, we trained for 1000 epochs before new data arrived and for 100 epochs after its arrival, in order to balance the total number of gradient updates before and after the introduction of new data.

## E.2 CONTINUAL PRETRAINING OF LLMS

For continual pretraining of LLMs, we report the results with 3 seeds.

Table 3: Detailed settings in continual pretraining of LLMs.

| Parameter | Value |
|---|---|
| Optimizer | AdamW (Loshchilov & Hutter, 2017) |
| Weight Decay | 1e−1 |
| Learning Rate | 6e−4 |
| Minimum Learning Rate | 6e−5 |
| Learning Rate Scheduler | Warmup + Linearly Decaying |
| Gradient norm clipping | 1.0 |
| Batch Size | 480 |

We used implementation and hyperparameters of nanoGPT from Karpathy (2023).

During first phase, the learning rate linearly increases from 0 to the target learning rate (6e−4) during 2,000 steps, then annealed to minimum learning rate (6e−5) until the end of the phase. In the second phase, the learning rate linearly increases from 0 to the target learning rate (6e−4) during 10% of training iterations of second phase. Then, it decreases linearly to minimum learning rate (6e−5) until the end of the phase.

## E.3 REINFORCEMENT LEARNING

For reinforcement learning, we report the results with 5 seeds.

For S&P method, we apply S&P to the encoder and Reset to the fully connected layers (D'Oro et al., 2022) for discrete control, and S&P with $\lambda = 0.8$ to whole parameters for continuous control tasks. We perform a single intervention (Full Reset, S&P, FIRE) at the midpoint of learning. We followed the hyperparameter configurations used in prior work (Sokar et al., 2023).

Table 4: Hyperparameters used in the ALE environment with DQN algorithm.

| Parameter | Value |
|---|---|
| Optimizer | Adam (Kingma & Ba, 2014) |
| Optimizer: $\epsilon$ | $1.5e-4$ |
| Optimizer: Learning rate | $6.25e-5$ |
| Minimum $\epsilon$ for training | 0.01 |
| Evaluation $\epsilon$ | 0.001 |
| Discount factor $\gamma$ | 0.99 |
| Replay buffer size | $10^6$ |
| Minibatch size | 32 |
| Initial collect steps | 20000 |
| Training iterations | 10 |
| Training environment steps per iteration | $250K$ |
| Updates per environment step (Replay Ratio) | 1 |
| Target network update period | 2000 |
| Loss function | Huber Loss (Huber, 1992) |

For continuous tasks, the hyperparameter setting is followed by Lee et al. (2024b).

Table 5: Hyperparameters used in HumanoidBench environments with SimBa.

| Parameter | Value |
|---|---|
| Optimizer | AdamW (Loshchilov & Hutter, 2017) |
| Optimizer: Learning rate | $1e-4$ |
| Optimizer: Weight decay | 0.01 |
| Actor hidden dim | 128 |
| Actor num blocks | 1 |
| Critic hidden dim | 512 |
| Critic num blocks | 2 |
| Discount factor $\gamma$ | 0.99 |
| Clipped Double-Q (Fujimoto et al., 2018) | True |
| Replay buffer size | $10^6$ |
| Minibatch size | 256 |
| Initial collect steps | 5000 |
| Updates per environment step (Replay Ratio) | 2 |
| Soft target update factor $\tau$ | 0.005 |

### E.4 HYPERPARAMETER SEARCH SPACE

Table 6 presents the hyperparameter search space, and Tables 7-10 present their optimal values.

Table 6: Hyperparameter search space for all experiments.

| Experiment | Method | Hyperparameters | Search Space |
|---|---|---|---|
| Warm-Starting | S&P | $\lambda$ | $0.2, 0.4, 0.6, 0.8$ |
| | DASH | $\alpha$ | $0.1, 0.3$ |
| | | $\lambda$ | $0.05, 0.1, 0.3$ |
| | Parseval Reg. | $\lambda$ | $1e{-}3, 1e{-}4, 1e{-}5$ |
| | CBP | $\rho$ | $1e{-}4, 1e{-}5$ |
| | | $m$ | $100, 1000$ |
| | ReDo | $\tau$ | $0.01, 0.05, 0.1, 0.5$ |
| | L2 Init | $\lambda$ | $1e{-}3, 1e{-}4, 1e{-}5$ |
| | SNR | $\tau$ | $0.01, 0.02, 0.04, 0.08$ |
| Continual Learning | S&P | $\lambda$ | $0.2, 0.4, 0.6, 0.8$ |
| | DASH | $\alpha$ | $0.1, 0.3$ |
| | | $\lambda$ | $0.05, 0.1, 0.3$ |
| | Parseval Reg. | $\lambda$ | $1e{-}3, 1e{-}4, 1e{-}5$ |
| | CBP | $\rho$ | $1e{-}4, 1e{-}5$ |
| | | $m$ | $100, 1000$ |
| | ReDo | $\tau$ | $0.01, 0.05, 0.1, 0.5$ |
| | L2 Init | $\lambda$ | $1e{-}3, 1e{-}4, 1e{-}5$ |
| | SNR | $\tau$ | $0.01, 0.02, 0.04, 0.08$ |
| Class-Incremental Learning | S&P | $\lambda$ | $0.2, 0.4, 0.6, 0.8$ |
| | DASH | $\alpha$ | $0.1, 0.3$ |
| | | $\lambda$ | $0.05, 0.1, 0.3$ |
| | Parseval Reg. | $\lambda$ | $1e{-}3, 1e{-}4, 1e{-}5$ |
| | CBP | $\rho$ | $1e{-}4, 1e{-}5$ |
| | | $m$ | $100, 1000$ |
| | ReDo | $\tau$ | $0.01, 0.05, 0.1, 0.5$ |
| | L2 Init | $\lambda$ | $1e{-}3, 1e{-}4, 1e{-}5$ |
| | SNR | $\tau$ | $0.01, 0.02, 0.04, 0.08$ |
| Continual pretraining of GPT-0.1B | S&P | $\lambda$ | $0.2, 0.5, 0.8$ |

Table 7: Hyperparameters for Warm-Start setting.

| Dataset | Method | Value |
|---|---|---|
| CIFAR-10 (ResNet-18) | S&P | $\lambda = 0.8$ |
| | DASH | $\alpha = 0.3, \lambda = 0.05$ |
| | Parseval Reg. | $\lambda = 1e{-}3$ |
| | CBP | $\tau = 1e{-}4, m = 1000$ |
| | ReDo | $\tau = 0.5$ |
| | L2 Init | $\lambda = 1e{-}3$ |
| | SNR | $\tau = 0.01$ |
| | FIRE | iter $= 10$ |
| CIFAR-100 (ViT-Tiny) | S&P | $\lambda = 0.8$ |
| | DASH | $\alpha = 0.1, \lambda = 0.05$ |
| | Parseval Reg. | $\lambda = 1e{-}5$ |
| | CBP | $\tau = 1e{-}4, m = 100$ |
| | ReDo | $\tau = 0.5$ |
| | L2 Init | $\lambda = 1e{-}3$ |
| | SNR | $\tau = 0.01$ |
| | FIRE | iter $= 10$ |
| Tiny ImageNet (VGG-16) | S&P | $\lambda = 0.8$ |
| | DASH | $\alpha = 0.1, \lambda = 0.05$ |
| | Parseval Reg. | $\lambda = 1e{-}3$ |
| | CBP | $\tau = 1e{-}4, m = 1000$ |
| | ReDo | $\tau = 0.5$ |
| | L2 Init | $\lambda = 1e{-}3$ |
| | SNR | $\tau = 0.08$ |
| | FIRE | iter $= 10$ |

Table 8: Hyperparameters for Continual Setting.

| Dataset | Method | Value |
|---|---|---|
| CIFAR-10 (ResNet-18) | S&P | $\lambda = 0.8$ |
| | DASH | $\alpha = 0.3, \lambda = 0.05$ |
| | Parseval Reg. | $\lambda = 1e-3$ |
| | CBP | $\tau = 1e-5, m = 1000$ |
| | ReDo | $\tau = 0.05$ |
| | L2 Init | $\lambda = 1e-5$ |
| | SNR | $\tau = 0.08$ |
| | FIRE | iter $= 10$ |
| CIFAR-100 (ViT-Tiny) | S&P | $\lambda = 0.8$ |
| | DASH | $\alpha = 0.1, \lambda = 0.05$ |
| | Parseval Reg. | $\lambda = 1e-5$ |
| | CBP | $\tau = 1e-5, m = 1000$ |
| | ReDo | $\tau = 0.01$ |
| | L2 Init | $\lambda = 1e-4$ |
| | SNR | $\tau = 0.04$ |
| | FIRE | iter $= 10$ |
| Tiny ImageNet (VGG-16) | S&P | $\lambda = 0.8$ |
| | DASH | $\alpha = 0.1, \lambda = 0.1$ |
| | Parseval Reg. | $\lambda = 1e-4$ |
| | CBP | $\tau = 1e-5, m = 1000$ |
| | ReDo | $\tau = 0.01$ |
| | L2 Init | $\lambda = 1e-5$ |
| | SNR | $\tau = 0.01$ |
| | FIRE | iter $= 10$ |

Table 9: Hyperparameters for Class-Incremental Setting.

| Dataset | Method | Value |
|---|---|---|
| CIFAR-100 (ViT-Tiny) | S&P | $\lambda = 0.8$ |
| | DASH | $\alpha = 0.3, \lambda = 0.3$ |
| | Parseval Reg. | $\lambda = 1e-5$ |
| | CBP | $\tau = 1e-5, m = 1000$ |
| | ReDo | $\tau = 0.01$ |
| | L2 Init | $\lambda = 1e-5$ |
| | SNR | $\tau = 0.08$ |
| | FIRE | iter $= 10$ |
| Tiny ImageNet (VGG-16) | S&P | $\lambda = 0.8$ |
| | DASH | $\alpha = 0.3, \lambda = 0.1$ |
| | Parseval Reg. | $\lambda = 1e-3$ |
| | CBP | $\tau = 1e-4, m = 1000$ |
| | ReDo | $\tau = 0.05$ |
| | L2 Init | $\lambda = 1e-4$ |
| | SNR | $\tau = 0.02$ |
| | FIRE | iter $= 10$ |

Table 10: Hyperparameters for Continual pretraining of LLMs.

| Method | CKPT | Value |
|--------|------|-------|
| S&P    | Best | $\lambda = 0.5$ |
|        | 30k  | $\lambda = 0.8$ |
|        | 60k  | $\lambda = 0.5$ |
| FIRE   | Best | iter $= 5$ |
|        | 30k  | iter $= 5$ |
|        | 60k  | iter $= 5$ |

