# OpenReview forum: "FIRE: Frobenius-Isometry Reinitialization for Balancing the Stability–Plasticity Tradeoff"
_ICLR.cc/2026/Conference — ICLR 2026 Oral_

### Official Review · Reviewer_9wQe · 2025-11-01

**Soundness:** 3
**Presentation:** 3
**Contribution:** 3
**Rating:** 6
**Confidence:** 4

**Summary:**

The authors introduce FIRE which views resetting a network in continual learning as a constrained projection that keeps weights close to the previous solution (minimize squared Frobenius error) while restoring isometry (zero deviation-from-isometry), yielding the nearest orthogonal weights via an Orthogonal Procrustes step implemented efficiently with a few Newton–Schulz iterations. This method is motivated by the need to balance plasticity and stability. Theoretically, the authors demonstrate that this limits representation drift, smooths the loss (via a Hessian bound), raises effective rank, and bounds neuron dormancy, thus improving plasticity without sacrificing stability. Over a series of continual learning experiments in vision, language, and RL, the authors show that using FIRE at task boundaries (or once mid-training in RL) consistently maintains plasticity, attaining similar or superior performance to the benchmark methods that the authors evaluate.

**Strengths:**

- The paper is generally well written.
- The paper utilizes a good mix of continual learning benchmark problems: vision, language, and RL along with a good mix of network architectures.
- The derivation of FIRE is very well done and posing the stability-plasticity tradeoff as a hard constraint optimization problem leads to an algorithm that does not necessitate an explicit $\lambda$ weight that requires the user to tune how much the method should prioritize stability vs plasticity. Many continual learning methods are sensitive to hyperparameter choice and require careful tuning, and outside the well structured academic benchmarks, it's not clear how robustly these methods will perform.
- The theory is solid and well justifies FIRE.

**Weaknesses:**

- While the experiments are thorough, the empirical conclusions would be more convincing had more competitor methods been evaluated. Given that FIRE is partially motivated by parameter resets, it would be useful to compare FIRE against reset methods such as Self-Normalized Resets, Continual Backdrop, and ReDO which have been shown to be effective at maintaining plasticity and against the regularization based method of L2 Init.
- Tangential to the above point, FIRE appears to show similar performance to its competitor methods on the class incremental and data incremental settings. While in the warm start setting FIRE does perform the best, given the scale the performance in marginally better. Had additional baseline methods been evaluated it is not clear if FIRE would be consistently outperforming these baselines.
- While test error is the metric of interest, many papers on plasticity loss consider trainability and minimizing training error as a diagnostic of plasticity loss. It would be beneficial to also report training error for the experiments as plasticity loss and generalization error can confound with regularization and the number of training epochs, see appendix A.2 in Self Normalized Resets for Plasticity in Continual Learning.
- While FIRE is motivating by preserving stability, I don't quite see how the vision and language experiments evaluate stability, these look to me to be purely problems that evaluate plasticity. For instance in the vision problems, the network sees progressively more data, whether it is warm starting or class/data incremental. For the language problem, correct me if I am mistaken, but if the validation set is the second training set, then again I do not see how stability is measured.
- Additionally for the language problem, I don't quite see how plasticity is present. It appears that every method is able to reduce the validation error, and a plateau begins to appear roughly at the same iteration count for each method. It would be helpful, for instance, to see an experiment where the baseline's error (training or test) increases over time.

**Questions:**

- Given the noted similarity with Parseval, could the authors compare and contrast their contributions with that of the Parseval paper?
- What are the with reset and without reset dashed lines in row (a) of Figure 2?
- Have the author's found any limitations to FIRE? For instance, many continual learning methods can be sensitive to the choice of hyper parameter. Could there be data streams that could be particularly challenging for FIRE to train on?

---

> ### Author Response · Authors · 2025-11-26
> **Rebuttal (1/2)**
>
> > - While the experiments are thorough, the empirical conclusions would be more convincing had more competitor methods been evaluated. Given that FIRE is partially motivated by parameter resets, it would be useful to compare FIRE against reset methods such as Self-Normalized Resets, Continual Backdrop, and ReDO which have been shown to be effective at maintaining plasticity and against the regularization based method of L2 Init.
> > - Tangential to the above point, FIRE appears to show similar performance to its competitor methods on the class incremental and data incremental settings. While in the warm start setting FIRE does perform the best, given the scale the performance in marginally better. Had additional baseline methods been evaluated it is not clear if FIRE would be consistently outperforming these baselines.
>
> Thank you for your suggestion. We newly added **Continual Backprop, Self-Normalized Resets, ReDo, L2init and Muon** as our baseline method for continual visual learning. The updated results are shown in Figure (2). The result clearly shows that FIRE outperforms all newly added baselines.
>
>
> > While test error is the metric of interest, many papers on plasticity loss consider trainability and minimizing training error as a diagnostic of plasticity loss. It would be beneficial to also report training error for the experiments as plasticity loss and generalization error can confound with regularization and the number of training epochs, see appendix A.2 in Self Normalized Resets for Plasticity in Continual Learning.
>
> Thank you for pointing out this. We agree with your perspective on confounding factors. Thus, we updated our manuscript to provided train accuracies of our experiments in Appendix B.4. As conssistent with previous works [1, 2], all of methods achieve near perfect train accuracy.
>
> > - While FIRE is motivating by preserving stability, I don't quite see how the vision and language experiments evaluate stability, these look to me to be purely problems that evaluate plasticity. For instance in the vision problems, the network sees progressively more data, whether it is warm starting or class/data incremental. For the language problem, correct me if I am mistaken, but if the validation set is the second training set, then again I do not see how stability is measured.
> > - Additionally for the language problem, I don't quite see how plasticity is present. It appears that every method is able to reduce the validation error, and a plateau begins to appear roughly at the same iteration count for each method. It would be helpful, for instance, to see an experiment where the baseline's error (training or test) increases over time.
>
> In continual visual learning and reinforcement learning, stability is represented by performance drop after reinitialization. In Figure 2b and 2c, FIRE shows no performance drop after reinitialization, while full reset or DASH show clear drop.
>
> In continual pretraining of LLMs, the problem inherently requires both sufficient stability and plasticity to achieve optimal performance. Full-reset, which have perfect plasticity with zero stability, shows poor performance in this setting (see Figure 4). This indicate that mainting stability is critical to achieve good performance in this setting. Consistent with this observation, S&P and FIRE which provide a certain level of both stability and plasticity, achieve better performance than both the base model and the full-reset model.

---

> > ### Author Response · Authors · 2025-11-26
> > **Rebuttal (2/2)**
> >
> > > Given the noted similarity with Parseval, could the authors compare and contrast their contributions with that of the Parseval paper?
> >
> > Our contributions are differ from that of Parseval paper [3] in three key points.
> >
> > 1. **Theoretical analysis:** While [3] only empirically demonstrates that regularizing DfI improves plasticity, we provide a theoretical connection between previous plasticity measures (loss curvature, effective rank, dormant neurons) and DfI. This analysis offers a solid justification for minimizing DfI to enhance plasticity.
> > 2. **Experimental scope:** [3] considers only continual reinforcement learning, which is somewhat limited in scope. In contrast, we conduct extensive evaluations across diverse domains, including vision, language, and RL.
> > 3. **Convergence speed:** Parseval regularization [3] has a clear limitation: the regularization term interferes with the task gradient, leading to slower convergence. In contrast, our method, FIRE, does not suffer from this issue. The results in Figure 2a clearly show that Parseval regularization converges more slowly than our approach.
> >
> > > What are the with reset and without reset dashed lines in row (a) of Figure 2?
> >
> > In Figure 2a, **w/o warm start** is equivalent to **Full reset**, and **warm start** is equivalent to **Base**.
> >
> > > Have the author's found any limitations to FIRE? For instance, many continual learning methods can be sensitive to the choice of hyper parameter. Could there be data streams that could be particularly challenging for FIRE to train on?
> >
> > Since FIRE is a reinitialization-based method, it requires task boundary information. Therefore, scenarios in which task boundaries are not accessible are particularly challenging for FIRE. For example, in RL, the distribution changes gradually, making it difficult to determine the optimal reinitialisation point. Investigating the optimal reinitialization interval would be valuable future work.
> >
> > **References**
> >
> > [1] Ash, Jordan, and Ryan P. Adams. "On warm-starting neural network training." Advances in neural information processing systems 33 (2020): 3884-3894.
> >
> > [2] Lee, Hojoon, et al. "Slow and steady wins the race: Maintaining plasticity with hare and tortoise networks." arXiv preprint arXiv:2406.02596 (2024).
> >
> > [3] Chung, Wesley, et al. "Parseval regularization for continual reinforcement learning." Advances in Neural Information Processing Systems 37 (2024): 127937-127967.

---

### Official Review · Reviewer_iykC · 2025-11-01

**Soundness:** 3
**Presentation:** 3
**Contribution:** 3
**Rating:** 6
**Confidence:** 3

**Summary:**

The paper presents FIRE, a novel and principled method for reinitializing neural network weights to balance stability and plasticity in continual learning scenarios. The core idea is to formulate reinitialization as a constrained optimization problem: minimizing the Squared Frobenius Error (SFE) to the previous weights (stability) subject to the constraint of zero Deviation from Isometry (DfI) (plasticity). The solution is efficiently approximated via the Newton-Schulz iteration. The method is evaluated extensively across vision, language, and reinforcement learning tasks, demonstrating consistent superiority over naive training and baseline reinitialization methods like S&P and DASH.

**Strengths:**

- This paper introduces the novel use of Deviation from Isometry (DfI) as a unified and direct measure of plasticity loss, providing compelling theoretical evidence that minimizing DfI is mathematically equivalent to achieving multiple desirable plasticity properties, including flattening the loss surface, preventing neuron dormancy, and increasing feature rank.
- The solution, corresponding to the nearest orthogonal matrix via polar decomposition, is efficiently computed using the Newton–Schulz iteration, rendering the method practical with negligible computational overhead.
- The robustness of FIRE is convincingly demonstrated across diverse domains, including continual visual learning (ResNet, ViT), large language model pretraining (GPT-0.1B), and reinforcement learning (SAC, DQN), suggesting its broad applicability.

**Weaknesses:**

- The Newton–Schulz iteration used in FIRE is implemented with a fixed number of steps and lacks any adaptive stopping criterion or convergence check. For highly ill-conditioned weight matrices, the fixed step may be vulnerable to divergence, which could compromise the isometric constraint.
- While the theoretical claim of low time overhead is promising, providing concrete measurements would further strengthen its credibility. Additionally, a comparative analysis of memory footprint against baseline methods is lacking.
- The study does not examine the interaction of FIRE with other effective continual learning strategies, such as parameter-efficient fine-tuning (PEFT) methods like LoRA, which also target the stability and plasticity tradeoff. Besides, how does FIRE perform with other regularization or sharpness-aware landscape techniques?
- Which stage or task is evaluated in Figure 2c, and do the later tasks perform differently from the earlier ones?

**Questions:**

please ref to the weakness

---

> ### Author Response · Authors · 2025-11-26
> **Rebuttal**
>
> > The Newton–Schulz iteration used in FIRE is implemented with a fixed number of steps and lacks any adaptive stopping criterion or convergence check. For highly ill-conditioned weight matrices, the fixed step may be vulnerable to divergence, which could compromise the isometric constraint.
>
> We understand your concern regarding the Newton–Schulz (NS) iteration. However, in practice, divergence of NS iteration is extremely rare. First, in all of our experiments (Figures 2, 3, and 4), FIRE does not diverge. Second, our ablation study (Figure 5) demonstrates that FIRE successfully satisfies the isometric constraint (low DfI) without any signs of divergence. Third, the Muon optimizer [1], which also employs NS iteration, has been widely used in practice without reports of divergence issues.
> Taken together, these observations suggest that NS iteration is very unlikely to diverge in neural network settings.
>
> > While the theoretical claim of low time overhead is promising, providing concrete measurements would further strengthen its credibility. Additionally, a comparative analysis of memory footprint against baseline methods is lacking.
>
> Based on your suggestion, we report the wall-clock time and memory footprint during reinitialization and compare them with previous reinitialization methods.
>
> |Method|Wall-Clock Time|GPU Memory|
> |-|-|-|
> |Shrink Perturb|0.002 sec|27 MB  |
> |FIRE          |0.06 sec |55 MB  |
> |DASH          |69 sec   |2834 MB|
>
>
> As shown in the table, FIRE introduces negligible computational cost and memory usage. We updated our manuscript to include this result (Appendix B.1).
>
> > The study does not examine the interaction of FIRE with other effective continual learning strategies, such as parameter-efficient fine-tuning (PEFT) methods like LoRA, which also target the stability and plasticity tradeoff.
>
> Thank you for your insightful comment. Based on your suggestion, we have added Plasticity Injection (PI) [2], which is a PEFT method that shown to well balances stability and plasticity, as a baseline method for our RL experiment. While PI is good at maintaining stability, it shows poor plasticity. These results suggest that, FIRE can provide a more favorable stability–plasticity trade-off than the considered PEFT baseline.
>
> > Besides, how does FIRE perform with other regularization or sharpness-aware landscape techniques?
>
> Thank you for your suggestion. We agree that investigating the synergy between FIRE and other regularization or optimization methods would be a valuable future direction. However, since the scope of our work is to present and evaluate FIRE, we leave a detailed investigation of such synergies to future work.
>
> > Which stage or task is evaluated in Figure 2c
>
> We reported the full-task accuracy in Figure 2c. Here, “full-task accuracy” means that the test accuracy is evaluated on the entire dataset containing all classes.
>
> > do the later tasks perform differently from the earlier ones?
>
> In class-incremental setting (Figure 2c), the number of classes are continaully expanded (e.g. 5->10->25->...->100) and corresponding data for each class are added to the training set. Therefore later tasks differ from earlier tasks that they include more classes. Note that this setting is widely adopted in related literatures [3, 4].
>
> **References**
>
> [1] Jordan, Keller, et al. "Muon: An optimizer for hidden layers in neural networks, 2024."
>
> [2] Nikishin, Evgenii, et al. "Deep reinforcement learning with plasticity injection." Advances in Neural Information Processing Systems 36 (2023): 37142-37159.
>
> [3] Lewandowski, Alex, et al. "Learning Continually by Spectral Regularization." The Thirteenth International Conference on Learning Representations.
>
> [4] Park, Sangyeon, et al. "Activation by Interval-wise Dropout: A Simple Way to Prevent Neural Networks from Plasticity Loss." Forty-second International Conference on Machine Learning.

---

### Official Review · Reviewer_Ts7T · 2025-11-04

**Soundness:** 2
**Presentation:** 3
**Contribution:** 3
**Rating:** 6
**Confidence:** 4

**Summary:**

In this paper, the authors address the challenge of balancing plasticity and stability in continual learning by proposing a reinitialization-based method called FIRE. This approach is grounded in theoretical analysis and demonstrates its effectiveness across various learning paradigms, including continual visual learning, language modeling, and reinforcement learning.

**Strengths:**

The proposed method is strongly motivated and comes with theoretical guarantees (although I have not thoroughly checked the correctness of the proofs). Moreover, the authors validate the effectiveness of their approach across multiple learning paradigms, demonstrating its broad applicability through extensive experiments.

**Weaknesses:**

the authors only state the theorems without providing detailed analysis of the theorems and their corresponding implications, especially for Theorem 1 and Theorem 4. In addition, the choice of baseline methods for comparison is rather limited, which to some extent reduces the strength of the empirical validation of the proposed approach.

**Questions:**

In Theorem 1, it can be directly inferred that minimizing the SFE leads to a reduction in the variance difference. However, a more specific analysis is needed to explain how a smaller variance difference translates into strong stability for the model. For Theorem 4, the upper bound of $s_j$ is 1, but since $1<\sqrt{\frac{1+\epsilon}{1-\epsilon}}$ always holds, the derivation of the upper bound does not provide any meaningful insight. Meanwhile, the lower bound of $s_j$  increases as $\epsilon$ decreases, which actually implies an increase in the number of dormant neurons—contradicting the paper’s claim that the number of dormant neurons is reduced. Moreover, in Theorems 3 and 4, the assumption $\epsilon<1$ appears rather strong,  and its validity in practice should be justified. Additionally, the authors should provide a derivation for Equation (5).

---

> ### Author Response · Authors · 2025-11-26
> **Rebuttal**
>
> Thank you for your constructive comments. These are our answer to your comments.
>
> > the authors only state the theorems without providing detailed analysis of the theorems and their corresponding implications, especially for Theorem 1 and Theorem 4.
>
> Thank you for pointing this out. We updated our manuscript accordingly. We have added anaysis and implication of Theorem 1 and 4 in Section 3.1 and 3.2.
>
> The implication of Theorem 1 is that, the discrepancy in normalized feature covariances between two networks is primarily dominated by SFE and spectral norms of the weight matrices. Minimizing SFE is therefore an effective way to preserve feature similarity, especially when spectral norms are large.
>
> Theorem 4 have two implications:
> 1. To reduce dormant neurons it is critical to reduce the discrepancy in activations across neurons rather than uniformly scaling them. This is because the neuron activity score $s_j$ is normalized by their average.
> 2. Minimizing DfI tightens both the lower and upper bounds on $s_j$, thereby limiting the score discrepancy between neurons and effectively reduces dormant neurons.
>
> > In addition, the choice of baseline methods for comparison is rather limited, which to some extent reduces the strength of the empirical validation of the proposed approach.
>
> We understand your concern on limited baselines. Therefore, we newly added **Continual Backprop, Self-Normalized Resets, ReDo, L2init and Muon** as our baseline method for continual visual learning. The updated results are shown in Figure (2).
>
> > In Theorem 1, it can be directly inferred that minimizing the SFE leads to a reduction in the variance difference. However, a more specific analysis is needed to explain how a smaller variance difference translates into strong stability for the model.
>
> Normalized feature covariance (the variance difference) represents discrepancy between feature representations of two networks. Therefore small variance difference indicates that two networks have similar feature representation, which also means that it is likely for them to have similar empirical performance. Therefore, the small variance difference represents stability of the network.
>
> > For Theorem 4, the upper bound of $s_j$ is 1, but since $1 < \sqrt\frac{1+\epsilon}{1-\epsilon}$ always holds, the derivation of the upper bound does not provide any meaningful insight.
>
> We would like to correct the misconception: upper bound of $s_j$ is not 1. If we look at the definition of $s_j$, the denominator ${\frac{1}{b}\sum_{k=1}^b \mathbb{E}_{z}[|\sigma(\langle z, w_k \rangle)|]}$ is average of all activations, not sum. Note that we employed the original definition by [1].
>
> > Meanwhile, the lower bound of $s_j$ increases as $\epsilon$ decreases, which actually implies an increase in the number of dormant neurons—contradicting the paper’s claim that the number of dormant neurons is reduced.
>
> We would like to correct the misconception: increase of $s_j$ indicates reduce in dormant neurons. Sokar et al. [1] classified neurons with $s_j < \tau$ as dormant neurons. Hence, to reduce the number of dormant neurons, $s_j$ should be increased. It seems like the name "dormancy score" made this confusion. Thus, we renamed $s_j$ as "neuron activity score" and updated our manuscript accordingly.
>
> > Moreover, in Theorems 3 and 4, the assumption $\epsilon < 1$ appears rather strong, and its validity in practice should be justified.
>
> We acknowledge that the assumption is strong. It is a limitation of our theoretical analysis. However, we would like to emphasize that our analysis is the first attempt to theoretically link data-dependent and non-differentiable plasticity measures (loss curvature, effective rank, dormant neurons) to data-independent and differentiable measure (DfI). Our work build a foundation for handling theoretical perspective of plasticity in much easier and tractable manner. We leave enhancing practicality of the theory as a future work.
>
> > Additionally, the authors should provide a derivation for Equation (5).
>
> We updated Appendix A to include the derivation.
>
> **References**
>
> [1] Sokar, Ghada, et al. "The dormant neuron phenomenon in deep reinforcement learning." International Conference on Machine Learning. PMLR, 2023.

---

### Official Review · Reviewer_EWuU · 2025-11-07

**Soundness:** 3
**Presentation:** 4
**Contribution:** 3
**Rating:** 6
**Confidence:** 3

**Summary:**

This paper proposes the FIRE algorithm for training Neural networks continually that is able to better balance stability with plasticity. The way it does so by doing a constrained optimization of the Squared Frobenius Error (SFE) between the weights and past weights, subject to the Deviation from Isometry (DfI) property being 0. The DfI property measures how close the weights are to orthonormal, which has been linked to high plasticity, and the SFE ensures the weights don’t deviate far when trying to achieve the plasticity objective. The paper evaluates the algorithm across continual visual learning, warm starting LLM pretraining, and RL.

**Strengths:**

This is a clearly written paper and the algorithm is very intuitive. The results are fairly strong across a variety of domains, showing the generality of the idea. The suite of experiments and ablations is fairly comprehensive. The theoretical results in the provide a further justification for why the objective used (DfI) is appropriate for plasticity, as it addresses several of the issues commonly associated with loss of plasticity.

**Weaknesses:**

- Given the similarity of the DfI optimization to the Muon optimizer, I feel like the paper should have more of a discussion on the work beyond what they've provided. The paper claims that the difference is in the coefficients chosen for the Newton-Schulz iteration, and that muon favors speed over stability, but it is not clear if that would actually matter. It seems that muon would also be a good baseline for this paper.
- I have a few questions which I've outlined below.

**Questions:**

- Do you apply FIRE only at the start of distribution shifts, or is it a regular intervention? In RL it seems reinitialization only happens once, what happens when you reinitialize more often?
- I am assuming that before the reinitialization point, the different RL algorithms should be the same, so why are the results for the different algorithms different before the reinitialization point? If it’s just the result of randomness in the run, then maybe try an experiment where you take the same checkpoints/replay buffers and reinitialize them with the different strategies, rather than starting the runs from scratch. This would help disentangle the effect of what was potentially in the replay buffer with the effect of the reinitialization strategy.
- In Figure 5, what happens when you decrease the number of iterations even more? What is the breaking point?
- What is the type of change that happens with this reset? For example, on average, how much do the parameters move, is it generally a low loss path/is the reset to the same basin?
- This is lower priority, but I’d be curious to see how Hare and Tortoise does in your experiments as well, since it can also be thought of as a type of resetting baseline.

---

> ### Author Response · Authors · 2025-11-26
> **Rebuttal (1/2)**
>
> Thank you for your comments. This is our answer to the weaknesses and questions.
>
> > Given the similarity of the DfI optimization to the Muon optimizer, I feel like the paper should have more of a discussion on the work beyond what they've provided. The paper claims that the difference is in the coefficients chosen for the Newton-Schulz iteration, and that muon favors speed over stability, but it is not clear if that would actually matter.
>
> We acknowledge that the paper does not provide sufficient discussion of the relationship between Muon and FIRE, particularly regarding the choice of coefficients for the Newton–Schulz (NS) iteration. In response, we set FIRE’s coefficients to those of Muon and compared its performance against FIRE with the original coefficients. We did not observe any notable performance differences between the two configurations. The corresponding results are updated in Appendix B.3.
>
> > It seems that muon would also be a good baseline for this paper.
>
> Thank you for your suggestion. We added Muon as a baseline method in our continual visual learning experiments. We have updated Figure 2 to include the results. Muon optimizer gradually loses plasticity and shows poor final performance. This indicates that Muon is ineffective at mitigating plasticity loss.
>
> > Do you apply FIRE only at the start of distribution shifts, or is it a regular intervention? In RL it seems reinitialization only happens once, what happens when you reinitialize more often?
>
> We only apply FIRE only at the start of distribution shifts. In RL, we applied FIRE only once during training. The purpose of the RL experiment is to evaluate FIRE's ability to reduce plasticity loss in RL. Therefore, to avoid potential confounding factors and isolate the problem, we applied reinitialization only once.
>
> We have not tested reinitializing more than once during training, but we believe that investigating the effect of reinitialization frequency in RL is an important direction for future work. For example, applying FIRE too frequently may lead to instability, while applying it too infrequently may cause the agent to suffer from plasticity loss. Since distribution shifts in RL are typically continuous and smooth, it is difficult to determine an appropriate reinitialization schedule for RL agents. Therefore, it is very interesting future direction.
>
> > I am assuming that before the reinitialization point, the different RL algorithms should be the same, so why are the results for the different algorithms different before the reinitialization point? If it’s just the result of randomness in the run, then maybe try an experiment where you take the same checkpoints/replay buffers and reinitialize them with the different strategies, rather than starting the runs from scratch. This would help disentangle the effect of what was potentially in the replay buffer with the effect of the reinitialization strategy.
>
>
> Thank you for pointing this out. As you mentioned, the reason of performance difference before reinitialization is because of randomness in the run. Based on your suggestion, we re-run the RL experiment by using the same checkpoints when reinitializing the network. We obtained similar results with the checkpointing. The updated results are shown in Figure 4.
>
> > In Figure 5, what happens when you decrease the number of iterations even more? What is the breaking point?
>
> Thank you for pointing this out. To address your comment, we conducted an additional experiment. We found that even a single iteration substantially decreases DfI and is sufficient to yield a performance gain. However, we also observed that using too few iterations (fewer than 5) leads to poor stability. Empirically, SFE reaches its peak after the first iteration and then decreases as the iterations continue. In summary, using a small number of iterations (<5) can still provide sufficient performance benefits, but may suffer from instability and ultimately cause a performance drop. We have added these results in Appendix B.2.

---

> > ### Author Response · Authors · 2025-11-26
> > **Rebuttal (2/2)**
> >
> > > What is the type of change that happens with this reset? For example, on average, how much do the parameters move, is it generally a low loss path/is the reset to the same basin?
> >
> > The amount of parameter movement under FIRE is about 60% of that under full reset (see Figure 5b, right). We did not investigate mode connectivity between the original and modified weights. However, we believe that a detailed analysis of the optimization landscape changes induced by FIRE would be a very meaningful direction for future work. Our current finding regarding the optimization landscape is that FIRE reduces the sharpness of the loss landscape (see Figure 5b, left).
> >
> > > This is lower priority, but I’d be curious to see how Hare and Tortoise does in your experiments as well, since it can also be thought of as a type of resetting baseline.
> >
> > Thank you for your valuable suggestion. Unfortunately, due to limited computational resources, we are unable to include Hare and Tortoise as additional baselines in our experiments. We sincerely apologize that we are not able to incorporate them at this time.

---

### Author Response · Authors · 2025-12-04
**General Response**

We have significantly revised our manuscript and conducted additional experiments to address the reviewers' concerns regarding baseline comparisons, theoretical clarity, and the relationship with related works. Below is a summary of the key updates:

**1. Inclusion of Extensive Baselines (Addressing EWuU, Ts7T, 9wQe)**

A primary concern across multiple reviewers was the limited number of baseline comparisons. We have substantially expanded our evaluation to demonstrate FIRE’s superiority:
* **Continual Visual Learning (Fig 2):** We added **five** new baselines: *Continual Backprop, Self-Normalized Resets, ReDo, L2init,* and the *Muon optimizer*. FIRE consistently outperforms these methods in preserving plasticity and final accuracy.
* **Reinforcement Learning:** We added *Plasticity Injection (PI)* as a PEFT-based baseline. Results show PI maintains stability but suffers from poor plasticity compared to FIRE.

**2. Theoretical Clarifications and Proofs (Addressing Ts7T)**

We addressed the request for deeper theoretical analysis and derivation details:
* **Theorem Implications (Sections 3.1 & 3.2):** We added detailed explanations of Theorem 1 (linking SFE to feature covariance discrepancy) and Theorem 4 (showing that minimizing DfI tightens bounds on neuron activity).
* **Terminology Correction:** To resolve confusion regarding "dormant neurons," we renamed the "dormancy score" to **"neuron activity score" ($\rho$ in the manuscript)** and clarified that increasing $\rho$ reduces the number of dormant neurons.
* **Derivations:** We added the explicit derivation for Equation (5) in **Appendix A**.

**3. Relationship with Muon and Newton-Schulz Iteration (Addressing EWuU, iykC)**

We clarified the distinction between FIRE and the Muon optimizer:
* **Coefficients:** We demonstrated in **Appendix B.3** that adopting Muon’s coefficients does not change FIRE's performance, proving robustness.
* **Performance:** We showed that while Muon also uses Newton-Schulz iteration, it fails to mitigate plasticity loss effectively in continual learning settings compared to FIRE (updated **Fig 2**).
* **Stability:** We addressed concerns about the potential divergence of the Newton-Schulz iteration by clarifying that, empirically, no divergence was observed across any of our extensive experiments (Figures 2, 3, and 4). We further justified its stability by citing the widespread use of the Muon optimizer, which employs the same iteration without reported divergence issues.

**4. Experimental Rigor and Overhead (Addressing EWuU, iykC, 9wQe)**

* **RL Randomness:** We re-ran RL experiments using identical checkpoints to isolate the effect of reinitialization, confirming FIRE's performance gain is not due to random seed variation (**Fig 4**).
* **Computational Cost:** We added a wall-clock time and memory footprint analysis in **Appendix B.1**. FIRE is negligible in cost ($0.06$s, $55$MB) compared to methods like DASH ($69$s, $2834$MB).
* **Training Accuracy:** We added **Appendix B.4** to report training accuracy.

We believe these revisions comprehensively address the reviewers' feedback and strengthen the paper's empirical and theoretical contributions.

---

### Meta-Review · Area_Chair_8oLK · 2026-01-05

**Summary:**

This paper received four reviews, all with an initial rating of 6 (= marginally above the acceptance threshold). The authors provided a rebuttal to each of the four reviews, but unfortunately none of the four reviewers responded to this rebuttal before the premature end of the discussion phase.

All reviewers already supported acceptance of this paper in their initial review, and indeed no major concerns were raised in the reviews. As discussed in more detail below, my impression is that the minor concerns raised by reviewers EWuU, Ts7T and iykC are reasonably well addressed by the authors’ rebuttal, while for some concerns the authors – quite reasonably in my opinion – indicate that it is outside the scope of the current paper.

In my opinion, the main not-fully-addressed concern was raised by reviewer 9wQe. This concern is discussed in more detail below. However, I do not consider this a major or blocking issue. I think this issue could have been resolved had there been a full discussion phase, and I hope that the authors will still address this issue in the camera-ready version based on the more detailed comments below.

I agree with the reviewers that this is a strong paper that deserves to be accepted at ICLR. Moreover, given that the authors’ rebuttal substantially strengthened this paper, and that it is my expectation that it took away the most pressing remaining concerns of the reviewers (and I expect some of the reviewers would have probably raised their ratings had there been a full discussion phase), I recommend this paper for a spotlight or perhaps an oral presentation.

**Reviewer Concerns:**

In my opinion, the main not-fully-addressed concern was raised by reviewer 9wQe. This reviewer raised that the vision and language experiments do not really evaluate stability. The authors respond by saying: “In continual visual learning and reinforcement learning, stability is represented by performance drop after reinitialization. In Figure 2b and 2c, FIRE shows no performance drop after reinitialization, while full reset or DASH show clear drop.”

While it could indeed be argued these drops measure some kind of stability, I think it is rather unconvincing to have this as the sole measure of stability in the paper, both because these drops are somewhat hard to compare from the provided plots and because it is not clear why these drops are necessarily a problem or undesirable.

**Reviewer Scores:**

Reviewer EWuU raises several – in my opinion – mostly minor concerns and questions. The authors’ response to most questions seems quite satisfying, while to some questions the authors indicate it to be outside the scope of the paper. I expect this reviewer might have raised their rating to an 8.

Reviewer Ts7T asks the authors to provide some more explanations and derivations for several of the theoretical results. To the best of my judgement, the authors respond by providing most of what the reviewer asked for. I expect this reviewer might have raised their rating to an 8.

Reviewer iykC raises four quite specific, mostly minor weaknesses. The author rebuttal in response to these weaknesses is to-the-point and seems to address the raised concerns satisfactorily. I expect this reviewer might have raised their rating to an 8.

Reviewer 9wQe raises two important concerns. The first is about the relatively modest number of competitor methods that the authors had compared against. In response to this concern, the authors added several additional baselines after the rebuttal. (Although I have to say that I have not been able to thoroughly check the quality of the implementation of these baselines.) The second concern is about that the vision and language experiments do not really evaluate stability. See above for a more detailed discussion on this concern. My expectation is that after a full discussion phase this reviewer would probably still have supported acceptance, but I don’t know whether they would have raised their rating.

---

### Decision · Program_Chairs · 2026-01-26

Accept (Oral)